# MOGIC: Metadata-infused Oracle Guidance for Improved Extreme Classification

## Abstract

Retrieval-augmented classification and generation models significantly benefit from the *early-stage fusion* of high-quality text-based auxiliary metadata, often called memory, but they suffer from high inference latency and poor robustness to noise. In classifications tasks, particularly the extreme classification (XC) setting, where low latency is critical, existing methods incorporate metadata for context enrichment via an XC-based retriever and obtain the representations of the relevant memory items to perform *late-stage fusion* to achieve low latency. With an aim of achieving higher accuracy while meeting the low latency constraints, in this paper, we propose MOGIC, an approach for metadata-infused Oracle guidance for XC tasks. In particular, we train an early-fusion Oracle classifier with access to both query-side and label-side ground-truth metadata in the textual form. The Oracle is subsequently used to guide the training of any existing memory-based XC Disciple model via regularization. The MOGIC algorithm, when applied to memory-based XC Disciple models such as OAK, improves precision@1 and propensity-scored precision@1 by ∼2% on four standard datasets, at no additional inference-time costs to the Disciple model. We also show the feasibility of applying the MOGIC algorithm to improve the performance of state-of-the-art memory-free XC approaches such as NGAME or DEXA, demonstrating that the MOGIC algorithm can be used atop any existing XC-based approach in a *plug-and-play* manner. Finally, we also show the robustness of the MOGIC method to missing and noisy metadata settings. We will release code on acceptance.

## 1 Introduction

Context enrichment is the process of incorporating metadata from an external source (referred to as the **memory** for a model, comprising memory items) to improve the overall performance of a given task. While these approaches have gained popularity in generation, in the form of retrieval-augmented generation (or RAG) (Lewis et al., 2020; Gao et al., 2024; Fan et al., 2024), such context enrichment via an external store also benefits classification tasks (Guu et al., 2020a; Guo et al., 2023; Mohan et al., 2024; Long et al., 2022). This paper focuses on using memory to improve classification, and in particular when the label space is extremely large in the order of millions a.k.a. eXtreme Classification **(XC)** (Mohan et al., 2024). XC tasks involve sparse query representation, and are short-text in nature, dealing with matching queries to keywords (for sponsored search ads (Dahiya et al., 2021; Jain et al., 2016; Prabhu et al., 2018b)), queries to product titles (for product recommendations (Dahiya et al., 2021; Medini et al., 2019; Mittal et al., 2021)), or queries to webpage titles (for tagging (Babbar & Schölkopf, 2017; Chang et al., 2020; You et al., 2019)). In such XC tasks, auxiliary metadata often has relevant diverse information that the input query does not, which can be leveraged to provide better predictions. For example, on sponsored search ads task that involves query-to-ad-keyword prediction, the query-side metadata is obtained by mining the organic search webpage titles clicked in response to the query on the search engine, while on the Wikipedia categories prediction task, other Wikipedia article titles connected to the original page via hyperlinks could serve as the metadata. For a comprehensive overview of the XC preliminaries, please refer to Appendix L.

Memory based augmentation in XC remains an unsolved problem due to challenges of accurate and fast memory access (low latency requirements are imposed by XC). While the ideal/golden memory items linkages are typically available during training data (referred to as *ground-truth metadata*), at

Table 1: MOGIC (OAK), OAK, MOGIC (NGAME) & NGAME predictions; ground truth labels & Oracle predictions. Legend: Black (ground truth), Red (incorrect), Green (correct), Blue (missing)

| Query | Grass court | Tangbe |
|---|---|---|
| Ground truth labels | Clay court, Carpet court, Hardcourt | Mustang District, Kali Gandaki Gorge, Kali Gandaki River, Upper Mustang, Gandaki River |
| Ground-truth Query Metadata | Tennis terminology, Sports rules and regulations, Tennis court surfaces | Populated places in Mustang District |
| MOGIC (OAK) predictions | Clay court, Carpet court, Hardcourt, Video arcade, U.S. Men's Clay Court Championships | Mustang District, Kali Gandaki Gorge, Kali Gandaki River, Upper Mustang, Gandaki River |
| OAK predictions | Fernie Ghostriders, Garland, Texas, List of Nevada state prisons, Ronald Reagan Boyhood Home, West End (Richmond, Virginia) | Desalpur, Second Franco-Dahomean War, Vladivostok, Kitenge, List of currently erupting volcanoes |
| MOGIC (NGAME) predictions | Clay court, Carpet court, Hardcourt, Stadium, Riding hall | Mustang District, Upper Mustang, Mustang Caves, List of municipalities in Andhra Pradesh, Muktinath |
| NGAME predictions | National Register of Historic Places listings in Cumberland County, North Carolina, Shangri La (Doris Duke), Vauxhall, National Register of Historic Places listings in Perry County, Alabama | Five kings of Wa, Piteraq, Wemale, Oyo Empire, List of lighthouses in Togo |
| Oracle predictions | Clay court, Carpet court, Hardcourt, Plexicushion, DecoTurf | Mustang District, Kali Gandaki Gorge, Kali Gandaki River, Upper Mustang, Mustang Caves |

test time, these would be predicted from the available set (referred to as *predicted metadata*). Mohan et al. (2024) found that the predicted metadata for test-time queries may either be unavailable (poor zero-shot generalization), or the retrievals might be irrelevant (noisy metadata). These challenges can be characterized into: (a) *sensitivity to retrieved metadata*: low quality retrieval from memory leads to noisy augmentation to the query, degrading task performance. For example, Cuconasu et al. (2024); Yoran et al. (2024); Yu et al. (2023) have showed that, for text-based early-fusion models, robustness to retrieved memory item quality is critical to performance. (b) *latency dependence on form and timing of metadata infusion*: textual metadata offers higher interpretability than embedding based metadata, but textual metadata has high inference-time fusion latency. Various state-of-the-art approaches within the retrieval augmentation settings can be viewed as balancing trade-offs between these challenges.

A recent approach to leverage memory metadata in XC (Mohan et al., 2024; Chien et al., 2023) demonstrated the advantage of using metadata to improve online extreme classification, there are multiple shortcomings. Since these approaches employs late-stage embedding-based fusion, their model shows gains in terms of generalization and inference latency, but the quality of the representations could be sub-par compared to early-fusion models. Additionally, unique to the classification setting, there could also be label-side metadata that can be incorporated within the memory framework.

In this paper, we design a novel memory-based approach for extreme classification that utilizes both query and label-side metadata and maintains low inference latency, by employing a combination of early-fusion of text-based metadata and late-fusion of memory items. Our approach involves two phases of training: Oracle training and Oracle-guided Disciple training. In the first phase, we train an early-fusion *Oracle* classifier which has access to both query-side and label-side ground-truth metadata in the text form. In the second phase, the Oracle is used to guide the training of any existing memory-based XC Disciple model, by means of a regularization loss. Table 1 shows two examples of queries with ground truth labels; predicted and ground truth memory items; and predictions from MOGIC (OAK), OAK, MOGIC (NGAME), NGAME (Dahiya et al., 2023a) and Oracle. We observe that for the first example, "Courts by type" is a good memory item, but some predicted metadata ("Landforms" and "Grasslands") mislead OAK to produce bad predictions about geographical places; NGAME predictions are also bad. However, with MOGIC regularized training, MOGIC (OAK) and MOGIC (NGAME) were able to retain the original intent of the query and predicted various type tennis courts. Furthermore, in the second example, when retrieved metadata is completely irrelevant to the query "Tangbe" which is about a village in Nepal, OAK's prediction was completely around the wrongly retrieved metadata while after Oracle guidance, MOGIC (OAK) was able to ignore the noisy information and retrieved the right Wikipedia See Also pages of Tangbe village.

Overall, our contributions are:

- We propose **MOGIC**, a Metadata-infused Oracle Guidance framework for Improved Extreme Classification, that maintains real-world inference latency, while achieving 1-2% improvement over state-of-the-art XC models.

- In the first phase, we train an early-fusion *Oracle* classifier which has access to both query-side and label-side ground-truth metadata in the text form. In the second phase, the Oracle is used to guide the training of any existing memory-based XC Disciple model, such as OAK, by means of a regularization loss.

- Extensive experiments on four popular benchmark XC datasets show that (1) MOGIC improves accuracy significantly in terms of standard metrics like precision, NDCG and propensity scored precision atop both memory-based models like OAK as well as memory-free models like DEXA and NGAME. (2) MOGIC is robust to missing and noisy metadata compared to the Oracle. (3) MOGIC (OAK) gives state-of-the-art XC metrics across all four datasets.

## 2 RELATED WORK

**Extreme Classification (XC)**: XC is a crucial component in ranking and recommendation systems (You et al., 2019; Guo et al., 2019; Dahiya et al., 2021; Mittal et al., 2021; Saini et al., 2021; Gupta et al., 2023; Mohan et al., 2024). XC approaches learn a classifier associated with each of the classes in the multi-label setting, with features obtained via classical approaches such as bag-of-words (Babbar & Schölkopf, 2017; Prabhu et al., 2018b) or decision trees (Prabhu et al., 2018b) or deep-learning techniques that leverage either pre-trained (Jain et al., 2019) or learned (You et al., 2019; Jiang et al., 2021; Dahiya et al., 2023a) features. The closest approach to ours is that of OAK (Mohan et al., 2024), wherein an XC classifier, such as an NGAME (Dahiya et al., 2023a) encoder, is used to retrieve the metadata, and a single transformer attention layer is used to fuse the representations of both the query and the retrieved metadata. The proposed MOGIC algorithm leverages a text-based early-fusion model to improve the representations of the memory items in OAK. Along another direction, models such as DEXA (Dahiya et al., 2023b) aggregate information from the neighborhood of the encoder representations to form the context. Consequently, as we show in Table 7, the MOGIC algorithm can also be applied atop models such as NGAME and DEXA to improve performance.

**Retrieval-augmented Generation (RAG)**: The RAG paradigm has become the defacto approach for incorporating metadata for context enrichment in generative model, with the application typically being that of question answering. Prior to RAG, models such as REALM (Guu et al., 2020b) have leveraged external knowledge sources to improve the accuracy of the transformer encoders using a retriever that selects relevant documents or passages from the memory, while an encoder fuses the input text and memory items, computing an enriched embedding. RAG-based approaches (Lewis et al., 2020; Akyurek et al., 2023; Zhang et al., 2023; Radhakrishnan et al., 2024; Muennighoff et al., 2024) combine pre-trained parametric and non-parametric memory for language generation. In RAG settings, the memory, typically text-based, is infused with the query at input (Yang et al., 2018; Karpukhin et al., 2020; Qu et al., 2021; Lan et al., 2023; Lála et al., 2023; Yan et al., 2024). Other approaches incorporate task-specific memory, such as tabular data (Zha et al., 2023; Luo et al., 2023) or knowledge graphs (Gaur et al., 2022; He et al., 2024). We observe that retrieval-augmented models benefit from early-fusion, and high-quality metadata, but suffer from high inference latency and poor robustness to noise. MOGIC makes a novel contribution by introducing the textual early-fusion of metadata into XC models while respecting latency constraints.

**Guided Representation Learning**: Transferring capabilities via context-following from large language models (LLMs) to smaller ones (Kim & Rush, 2016; Gupta & Agrawal, 2022; Xu et al., 2024) has been widely studied. In the generative setting, models such as Alpaca (Taori et al., 2023), Vicuna (Chiang et al., 2023), Self-instruct (Wang et al., 2023), etc., have been shown to use supervised instruction-following fine-tuning to improve generation where the tuning data was generated using LLMs. On the LLM-based classification task, AugGPT (Dai et al., 2023) employs a teacher LLM to rephrase input sentence to improve general and clinical-domain classification performance. Various other works (Gilardi et al., 2023; He et al., 2023; Gao et al., 2023; Li et al., 2024; 2023) have considered guidance for LLM-based classification in the context of annotation generation, data clustering and curation, etc., but do not target the XC setting. The Oracle guidance framework in

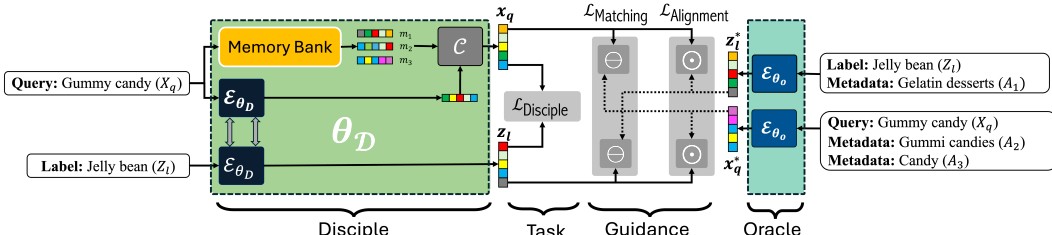

Figure 1: MOGIC robust training framework can be used with any XC or dense retrieval approach. In the given figure, MOGIC is used over OAK (Disciple) and its task specific loss. An Oracle's (LLaMA-2 or Phi-2 or DistilBERT) embeddings are used to regularize the representation of OAK using the guidance loss. The green box represents the OAK Disciple architecture (containing encoder, memory bank and combiner $\mathcal{C}$).

MOGIC can be viewed as an instantiation of guided representation learning, wherein the text-based early-fusion Oracle provides supervision for a downstream model such as OAK.

## 3 THE MOGIC APPROACH

**Notation:** Consider the task of query to label subset prediction, as common in the XC setting. Let $L$ be the total number of labels present, and $Q$, the total number of queries. Let $X_q, Z_l$ be the textual descriptions of the query (indexed by $q$) and label (indexed by $l$) respectively. For each query $X_q$, its ground truth label vector is $\mathbf{y}_q \in \{-1, +1\}^L$, where $y_{ql} = +1$ if label $l$ is relevant to the $q^{\text{th}}$ query and $y_{ql} = -1$ otherwise. Let $A_k$ be the textual descriptions of the memory item (indexed by $k$). For each query $X_q$ or label $Z_l$, its ground-truth memory-item vector is denoted by $\mathbf{a}_{q/l}$ or $\mathbf{a} \in \{-1, +1\}^M$, $M$ being the total number of memory items, where $a_k = +1$ if memory item $k$ is relevant otherwise $a_k = -1$.

In summary, $\mathcal{X} \stackrel{\text{def}}{=} \{X_q\}_{q=1}^Q \cup \{Z_l\}_{l=1}^L \cup \{A_k\}_{k=1}^M$ denote all the textual information, comprising $Q$ labeled queries, $L$ labels, $M$ memory items. The dataset is then denoted by $\{\{X_q, \mathbf{y}_q, \mathbf{m}_q\}_{q=1}^Q, \{Z_l, \mathbf{m}_l\}_{l=1}^L, \{A_k\}_{k=1}^M\}$. Within this setting, the XC problem is one of predicting labels $\tilde{\mathbf{y}}_q$ associated with each query $X_q$, while leveraging the memory items. We now present the MOGIC framework using these notations.

### 3.1 THE MOGIC FRAMEWORK

MOGIC comprises four main components, (a) The base XC model (Disciple), either memory-based, or memory free; (b) XC task specific loss function; (c) Guidance loss function and (d) The Oracle $\mathcal{O}$ for guidance. In this paper, we primarily focus on memory-based XC models, and in particular, OAK. Please refer to Figure 1 for overall understanding of the integration of OAK into the MOGIC framework. For more details, please refer to Appendix J. The four blocks are described below.

1. **Disciple**: Disciple, in this case, OAK is a trainable XC architecture with parameters $\theta_{\mathcal{D}}$ such that this model takes in query or label and outputs $d$ dimensional embedding such that $\mathcal{E}_\theta : \mathcal{X} \to \mathcal{S}^{d-1}$ lies on a unit sphere $\mathcal{S}^{d-1}$.

2. **Task-specific loss function**: This loss function denoted by $\mathcal{L}_{\text{Disciple}}$ is associated with the task for which the Disciple is being trained. For instance for OAK, $\mathcal{L}_{\text{Disciple}}$ is a triplet margin loss function described in (Mohan et al., 2024).

3. **Guidance loss function**: The guidance is passed on by the Oracle to the Disciple using the loss functions $\mathcal{L}_{\text{Alignment}}$ and $\mathcal{L}_{\text{Matching}}$, more details in Section 3.3.

4. **Oracle** ($\mathcal{O}$): Oracle is an encoder model (an LLM or a small language model (SLM)) with parameters $\theta_O$ which is used to guide the Disciple in the second training phase of MOGIC framework. A typical Oracle is computationally expensive but highly accurate embedding based model. The model takes in query as well as labels and its associated metadata as input to generate high quality representations in terms of the XC task. These embeddings are then used to guide the Disciple using Guidance loss. In MOGIC, we explore various

Oracle models such as DistilBERT (Sanh et al., 2019), LLaMA-2 (Touvron et al., 2023) and Phi-2 (Javaheripi et al., 2023) finetuned specifically for the XC task.

MOGIC is a highly modular framework which can incorporate different choices of the Disciple model and its task specific loss function. The following section discusses the training of the Oracle and development of the guidance loss to regularize the Disciple.

### 3.2 MOGIC PHASE 1: ORACLE TRAINING

To train a highly accurate Oracle XC model three components are critical: a) task-specific loss function, b) supervised training data and c) additional metadata which can enhance the textual quality of label and query. Much work has been done on designing effective task-specific loss functions (Dahiya et al., 2023a; Gupta et al., 2023; Kharbanda et al., 2023). We leverage standard triplet loss with in-batch negative sampling for training our Oracle model. The Oracle is trained by using an early fusion technique where query and label as well as their associate metadata is provided at input via simple text concatenation (with appropriate delimiter tokens). The input in then projected into an embedding space $\mathcal{R}^D$ using an encoder $\mathcal{E}_{\theta_O}$. For instance for a query $q$ with textual description $X_q$, the input to Oracle is simple text concatenation with its associated metadata ($m$ memory items), i.e., $\tilde{X}_q = X_q || M_{q_1} || \ldots || M_{q_m}$ and corresponding embedding is given as $\mathbf{x}_q^* = \mathcal{E}(\tilde{X}_q | \theta_O)$. Similarly for the label side, metadata rich label representation is computed as $\mathbf{z}_l^* = \mathcal{E}(\tilde{Z}_l | \theta_O)$ The optimization objective for Oracle (under the triplet loss) is given as follows.

$$\theta_O = \arg\min_\theta \mathcal{L}\Big( \{\mathbf{x}_q^*\}_{q \in Q}, \{\mathbf{z}_l^*\}_{l \in L}, \{y_{ql}\}_{q \in Q, l \in L} \Big) \tag{1}$$

where $\mathcal{L}$ can be any discriminative loss function that brings the relevant labels closer and pushes the irrelevant labels farther away from the query in their joint embedding space. Empirically, a triplet-loss based optimization is observed to give best model performance with dual-encoder based models:

$$\mathcal{L}_{\text{Triplet}}\Big( \{\mathbf{x}_q\}, \{\mathbf{z}_l\}, \{y_{ql}\} \Big) = \sum_{\substack{p:y_{qp}=+1 \\ n:y_{qn}=-1}} [\mathbf{x}_q^\top \mathbf{z}_n - \mathbf{x}_q^\top \mathbf{z}_p + \gamma]_+ \tag{2}$$

where $\mathbf{z}_p$ and $\mathbf{z}_n$ are positive and negative label embeddings and $\mathbf{x}_q$ is the query embedding corresponding to query $q$, and $\gamma$ is the margin. Note that subscripts in LHS have been omitted for notational simplicity.

For small language models, we perform LoRA finetuning for the specific XC task using the corresponding supervised training data. We use a simple prompt for both LLaMA-2 and Phi-2 as mentioned in the Appendix H.

### 3.3 MOGIC PHASE 2: ORACLE-GUIDED DISCIPLE TRAINING

An Oracle can demonstrate high accuracy on the downstream XC task due to its larger size or due to access to privileged information (ground-truth *textual* metadata) not accessible to the Disciple. However, they are computationally expensive to deploy, and have a high inference time, making them impractical for any real world applications. So, we use guidance in the form of embeddings from the Oracle to regularize a Disciple model of choice. Although we show experimental results using multiple Disciples in Section 4, in this section, we base our discussion around the OAK Disciple because it has been shown to be the state-of-the-art for XC tasks.

Now Disciple comes with two components: (1) an embedding generator which provides embeddings for a query $X_q$ and for a label $Z_l$ and (2) a task specific loss function over which Disciple was trained. Now, MOGIC proposed two additional loss terms namely `Alignment` loss and `Matching` loss to provide Oracle-guidance to Disciple and learn accurate embeddings. Both of these additional loss components is described below.

1. **Alignment**: This loss component focuses on aligning the ranking of the Oracle and the Disciple. To enforce this MOGIC introduce triplet margin loss between Oracle query and Disciple label embeddings and visa versa as follows.

$$\mathcal{L}_{\text{Alignment}} = \mathcal{L}_{\text{Triplet}}\Big( \{\mathbf{x}_q\}, \{\mathbf{z}_l^*\}, \{y_{ql}\} \Big) + \mathcal{L}_{\text{Triplet}}\Big( \{\mathbf{x}_q^*\}, \{\mathbf{z}_l\}, \{y_{ql}\} \Big)$$

2. **Matching**: This loss component focuses on ensuring that the Disciple mimics the Oracle's embeddings. To enforce this in MOGIC, we introduce mean squared error loss between Oracle query embeddings and student query embeddings, and similarly for labels as follows.

$$\mathcal{L}_{\texttt{Matching}} = \sum_{q \in Q} \left\| \mathbf{x}_q - \mathbf{x}_q^* \right\|_2 + \sum_{l \in L} \left\| \mathbf{z}_l - \mathbf{z}_l^* \right\|_2$$

where $\mathbf{x}_q = \mathcal{E}(X_q | \theta_D)$ and $\mathbf{z}_q = \mathcal{E}(Z_q | \theta_D)$ are embeddings corresponding to query $q$ and label $l$ from the Disciple. Finally, MOGIC combines the additional loss functions with the Disciple task-specific loss function and optimizes them simultaneously as shown below. Note that during the guidance training no gradient is passed to the Oracle model as shown in Figure 1.

$$\min_{\theta_D} \mathcal{L}_{\texttt{Disciple}} + \alpha \mathcal{L}_{\texttt{Alignment}} + \beta \mathcal{L}_{\texttt{Matching}}$$

where $\alpha, \beta$ are tunable hyper-parameters and set to $0.1$.

### 3.4 Theoretical Justification of Oracle-Guided Losses

The Oracle-guided training, where the Oracle is frozen and the Disciple is trained through `Alignment` and `Matching` losses, can lead to a significantly more accurate Disciple model owing to implicit knowledge transfer. Additionally, such guidance also leads to a more robust convergence of the Disciple's parameters in terms of the required training sample quantity. In this part, we theoretically demonstrate the effects of minimizing `Alignment` and `Matching` losses on the accuracy and sample complexity of the trained Disciple.

For the sake of analysis, we assume a canonically simplified problem setting. Specifically, we assume a Lipschitz continuous binary classification loss, with Lipschitz constant $K$, in place of the non-decomposable triplet loss. We also assume that all embeddings are bounded in norm by $B$. These assumptions are detailed in the Appendix A. The Disciple model is assumed to be a dual encoder with parameters $\theta_D = \{\theta_D^q, \theta_D^l\}$ where $\theta_D^q \in \mathcal{F}, \theta_D^l \in \mathcal{G}$ are the parameters on the query and label tower respectively. $\mathcal{F}, \mathcal{G}$ are the query and label-side hypothesis classes whose complexities are assumed to be bounded by Rademacher constants (Mohri & Talwalkar, 2018) $R_q, R_l$ respectively.

The following is our key theoretical result:

**Theorem 1.** *Given the problem setting described above, if the Disciple model is trained by minimizing the Oracle-guided loss $\mathcal{L} = \mathcal{L}_{\texttt{Alignment}} + KB \cdot \mathcal{L}_{\texttt{Matching}}$ on the training set $\mathcal{D} \sim D$ with $N$ samples, then for some $\lambda > 0$ and any $\delta \in [0, 1]$, the following inequality holds true with probability at least $1 - \delta$:*

$$\mathbb{E}_{((X,Z),y) \sim D} \mathcal{L}_{\texttt{Disciple}} \leq \mathbb{E}_{((X,Z),y) \sim D} \mathcal{L}_{\texttt{Oracle}} + \frac{4K}{N} \cdot (R_q + R_l) + 2\sqrt{\frac{\log(\frac{1}{\delta})}{N}} \tag{3}$$

*Proof.* Proof is provided in the Appendix A. □

The above theorem shows that, post Oracle-guided training, the Disciple's expected population loss tends to be close to Oracle's population loss itself, thus inheriting strong Oracle accuracy. Additionally, due to the separable training of $\theta_D^q, \theta_D^l$ parameters, there is no significant interactions between the two hypothesis classes thus leading to a smaller dependence on the complexities, *i.e.*, $R_q + R_l$. In contrast, a typical supervised training loss with joint optimization on the cross-product space $\theta_D^q \times \theta_D^l$ will require a much larger sample complexity. This implies good training efficiency for the proposed MOGIC framework.

## 4 Experiments and Results

### 4.1 Datasets and Experimental Setup

The XML Repository (Bhatia et al., 2016) provides various public XC datasets which are thoroughly studied and benchmarked by plethora of papers (You et al., 2019; Guo et al., 2019; Dahiya et al.,

Table 2: A summary of the dataset statistics in terms of the queries (Q), labels (L) and memory items (M). The Avg. queries per label is computed as the average value of the number of positive labels associated with each query in the dataset. Similarly, to find the Avg. labels per query, given a label, we identify the number of queries for which that label is a positive, and compute the average of those numbers. For WikiSeeAlso tasks, the Wikipedia categories that these articles are tagged with are used as metadata. For LF-WikiTitles-500K and LF-Wikipedia-500K tasks, the Wikipedia article titles connected to each original page via hyperlinks in the article are used as metadata.

| Dataset | #Train-Q | #L | #Test-Q | Avg. Q-per-L | Avg. L-per-Q | #M | Avg. M-per-Q |
|---|---|---|---|---|---|---|---|
| LF-WikiSeeAlsoTitles-320K
LF-WikiSeeAlso-320K | 693K | 312K | 177K | 2.11 | 4.67 | 656K | 4.89 |
| LF-WikiTitles-500K
LF-Wikipedia-500K | 1.8M | 501K | 783K | 4.74 | 17.15 | 2.1M | 15.95 |

2021; Mittal et al., 2021; Saini et al., 2021; Gupta et al., 2023; Mohan et al., 2024). But very few of them (Mohan et al., 2024; Chien et al., 2023) offer ground truth metadata. To fix this, we attach ground truth auxiliary data from the original dumps to existing XC datasets. Table 2 shows summary of dataset statistics. More details about the datasets are as follows.

The Wikipedia datasets are created from publicly available Wikipedia dumps[1]. The task in the **LF-WikiSeeAlsoTitles-320K** and **LF-WikiSeeAlso-320K** (full text version of the former) datasets is to, given a Wikipedia article/page, predict the other Wikipedia articles to be recommended in the 'See Also' section. The Wikipedia categories these articles are tagged with are used as metadata in this case. Similarly, **LF-WikiTitles-500K** and **LF-Wikipedia-500K** are datasets where the task is to, given a Wikipedia article/page, predict the Wikipedia categories the article should be tagged with. Other Wikipedia article titles connected to the original page via hyperlinks in the article are used as metadata in this case.

**Implementation details**: We initialize the encoder with a MS-MARCO (Chen et al., 2024) pre-trained DistilBERT and fine-tune it. Table 15 in the appendix summarizes all hyper-parameters for each dataset. Note that MOGIC uses golden linked metadata only at training time, whereas at inference time, these metadata linkages are induced (for a new query, links to metadata are predicted by the Disciple). MOGIC uses PyTorch and was trained on a machine with 4 AMD MI200 GPUs.

## 4.2 RESULTS

**Main results on benchmark datasets**: MOGIC is compared with state-of-the-art XC and dense retrieval approaches in Table 3. MOGIC leads to state-of-the-art accuracy on multiple datasets. These accuracy gains are attributed to gradient regularization from Oracle model.

In particular, MOGIC (OAK) outperforms OAK by 1-2% in P@1 and 2-3% in propensity scored metrics. In addition to OAK, MOGIC can even outperform graph based approaches like GraphFormers (Yang et al., 2021) and GraphSage (Hamilton et al., 2018) by 8%. These approaches are given an unfair advantage by providing ground truth memory items during inference. Note that MOGIC makes no change in input, only the training procedure is improved with an additional regularization loss. Also, MOGIC is simply a regularization framework and leads to no additional inference cost.

Figure 2 shows quantile wise-comparison of MOGIC and other methods for LF-WikiSeeAlsoTitles-320K and LF-WikiTitles-500K. The left-most bin contains the most rare/tail labels whereas the rightmost bin contains the most popular/head labels. MOGIC (OAK) gives consistent gains in tail bins and comparable results in head bins (see Appendix J for binning details).

**Impact of Oracle model**: MOGIC's improvement can be attributed to the Oracle model it uses to guide the gradients of the Disciple. In Table 4, we show results with three different models as Oracle: finetuned DistilBERT (which is our recommended choice of the Oracle), and two LoRA-finetuned SLMs, Phi-2 and LLaMa-2. We use PCA to downsize the embedding size of SLMs and use the resulting vectors to compute `Alignment` and `Matching` losses. Our recommended Oracle (DistilBERT), outperforms much larger Oracles (such as LLaMA-2 or Phi-2). For experiments replacing PCA with a linear projection layer, see Appendix F.

---

[1]https://dumps.wikimedia.org/enwiki/20220520/

Table 3: Main Results: Results on public benchmark datasets. MOGIC is up to 2% more accurate as compared to baselines. For details on evaluation metrics, see Appendix I. For results on LF-AmazonTitles-131K and LF-Amazon-131K datasets refer Appendix J.

| Method | P@1 | P@5 | N@5 | PSP@1 | PSP@5 | P@1 | P@5 | N@5 | PSP@1 | PSP@5 |
|---|---|---|---|---|---|---|---|---|---|---|
| | \multicolumn{5}{c}{LF-WikiSeeAlsoTitles-320K} | | | | | \multicolumn{5}{c}{LF-WikiTitles-500K} | | | | |
| MOGIC (OAK) | **34.62** | **17.93** | **27.44** | **35.70** | **33.18** | *47.28* | **18.55** | **34.97** | **27.29** | **26.12** |
| OAK | *33.71* | *17.12* | 24.53 | *33.83* | *30.83* | 44.82 | *17.67* | *33.72* | *25.79* | *24.90* |
| DEXA | 32.91 | 16.77 | 24.63 | 33.63 | 29.55 | **47.41** | 17.62 | 33.64 | 25.27 | 24.03 |
| NGAME | 32.64 | 16.60 | 23.44 | 33.21 | 29.87 | 39.04 | 16.08 | 30.75 | 23.12 | 23.03 |
| ANCE | 30.79 | 15.36 | *25.14* | 31.45 | 28.73 | 29.68 | 12.51 | 25.10 | 23.18 | 21.18 |
| DEXML | 29.90 | 14.80 | 22.80 | 30.70 | 25.70 | - | - | - | - | - |
| GraphFormers | 21.94 | 11.79 | 24.02 | 19.24 | 22.70 | 24.53 | 11.33 | 20.35 | 22.04 | 19.53 |
| GraphSAGE | 23.13 | 8.26 | 25.12 | 17.84 | 18.73 | 21.14 | 11.30 | 22.61 | 21.32 | 11.82 |
| | \multicolumn{5}{c}{LF-WikiSeeAlso-320K} | | | | | \multicolumn{5}{c}{LF-Wikipedia-500K} | | | | |
| MOGIC (OAK) | **49.62** | **24.26** | **50.49** | **36.15** | **43.17** | *85.34* | **51.50** | **77.85** | 43.60 | **61.74** |
| OAK | *48.57* | *23.28* | *49.16* | *33.92* | *40.44* | 85.23 | *50.79* | *77.26* | *45.28* | *60.80* |
| DEXA | 47.11 | 22.71 | 47.62 | 31.81 | 38.78 | 84.92 | 50.51 | 76.80 | 42.59 | 58.33 |
| NGAME | 46.40 | 18.05 | 46.64 | 28.18 | 33.33 | 84.01 | 49.97 | 75.97 | 41.25 | 57.04 |
| DEXML | - | - | - | - | - | **85.78** | 50.53 | 77.11 | - | 58.97 |
| PINA | 44.54 | 22.92 | - | - | - | 82.83 | 50.11 | - | - | - |
| ANCE | 45.64 | 17.32 | 45.43 | 29.60 | 32.83 | 77.92 | 40.95 | 68.70 | **50.99** | 57.33 |
| GraphFormers | 18.14 | 8.81 | 20.81 | 16.85 | 20.98 | 31.10 | 14.00 | 24.87 | 25.16 | 21.83 |
| GraphSAGE | 19.30 | 10.82 | 22.67 | 17.56 | 23.50 | 32.53 | 15.50 | 25.33 | 22.34 | 19.14 |

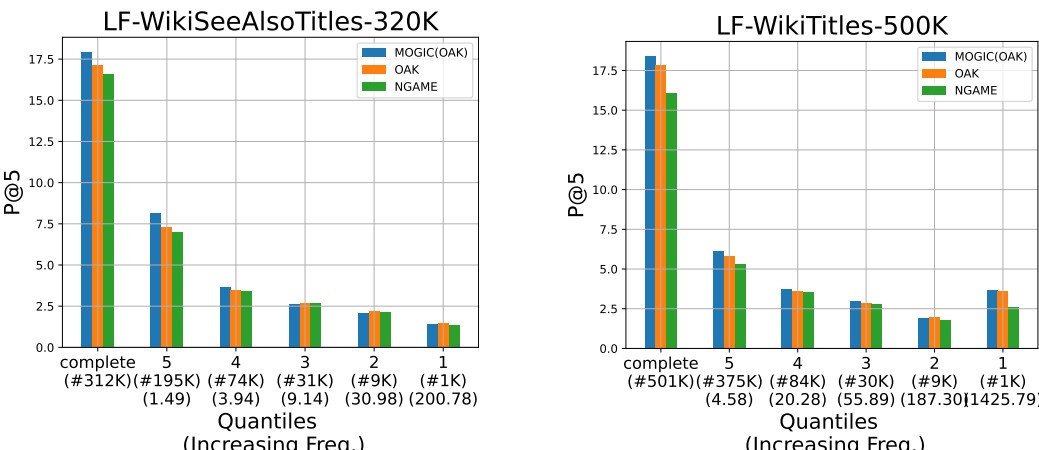

Figure 2: Quantile wise-comparison of MOGIC and other methods for LF-WikiSeeAlsoTitles-320K, LF-WikiTitles-500K

Table 4: Impact of Oracle models: MOGIC on the LF-WikiSeeAlsoTitles-320K dataset with different Oracle models. The DistilBERT Oracle outperforms the LoRA-finetuned SLM Oracles. Nevertheless, via the MOGIC framework, the Disciple (OAK) is capable of leveraging the Oracles' signals, even SLMs, to improve its task performance.

| Oracle Models | Finetuning | \multicolumn{5}{c}{MOGIC (OAK)} | | | | | \multicolumn{5}{c}{Oracle} | | | | |
|---|---|---|---|---|---|---|---|---|---|---|---|
| | | P@1 | P@5 | N@5 | PSP@1 | PSP@5 | P@1 | P@5 | N@5 | PSP@1 | PSP@5 |
| DistilBERT (65M params) | full | **34.62** | **17.93** | **27.44** | **35.70** | **33.18** | 42.78 | 20.53 | 32.99 | 43.59 | 37.57 |
| Phi-2 (2.7B params) | LoRA | *34.25* | *17.71* | *26.97* | *35.37* | *32.62* | 26.84 | 12.06 | 24.49 | 24.79 | 24.20 |
| LLaMA-2 (7B params) | LoRA | 33.94 | 17.43 | 26.87 | 34.92 | 32.10 | *29.57* | *13.40* | *26.69* | *27.38* | *26.74* |

**Ablations on Loss Functions**: To understand the importance of each loss function in the Phase 2 MOGIC training, we remove each of the 3 losses one by one. Lastly, we remove both the Oracle-guidance based loss functions (`Matching` and `Alignment`). Table 5 indicates all loss components are important for accurate results and the best accuracy is achieved when all components are used.

Table 5: Ablations on loss functions: We present ablations on `Alignment` and `Matching` losses. `Disciple + Alignment + Matching` is same as MOGIC (OAK).

| Loss terms in $\mathcal{L}$ | P@1 | P@5 | N@5 | PSP@1 | PSP@5 | P@1 | P@5 | N@5 | PSP@1 | PSP@5 |
|---|---|---|---|---|---|---|---|---|---|---|
| | | LF-WikiSeeAlsoTitles-320K | | | | | LF-WikiTitles-500K | | | |
| Disciple + Alignment + Matching | **34.62** | **17.93** | **27.44** | **35.70** | **33.18** | **47.28** | **18.55** | **34.97** | **27.29** | **26.12** |
| Disciple + Alignment | 34.12 | 17.66 | 26.72 | 35.16 | 32.57 | 45.22 | 17.58 | 33.49 | 27.24 | 25.10 |
| Disciple + Matching | 34.11 | 17.63 | 26.83 | 35.24 | 32.4 | 46.03 | 16.86 | 32.86 | 26.87 | 24.19 |
| Disciple | 33.71 | 17.12 | 24.53 | 33.83 | 30.83 | 44.82 | 17.67 | 33.72 | 25.79 | 24.90 |
| Alignment + Matching | 32.7 | 16.92 | 26.03 | 33.6 | 31.3 | 44.93 | 17.4 | 33.18 | 26.87 | 24.73 |

**Impact of using ground truth metadata**: A practical system uses metadata predicted by the Disciple given a query. Hence, all results so far for MOGIC have been reported based on predicted metadata linkages. In an ideal scenario, using ground truth metadata at test time could significantly boost overall XC task accuracy. In MOGIC we observe a large gap in performance of Oracle and MOGIC model, this is because Oracle model uses ground truth links to metadata during training and testing.

Table 6: Impact of using ground truth metadata: MOGIC is robust to noise but Oracle accuracy drops by 17% in P@1 on LF-WikiSeeAlsoTitles-320K.

| Models | Metadata Source | P@1 | P@5 | N@5 | PSP@1 | PSP@5 |
|---|---|---|---|---|---|---|
| MOGIC (OAK) | Ground-truth | **36.94** | **19.12** | **29.00** | **38.42** | **35.07** |
| | Predicted | 34.62 | 17.93 | 27.44 | 35.70 | 33.18 |
| Oracle | Ground-truth | 42.78 | 20.53 | 32.99 | 43.59 | 37.57 |
| | Predicted | 25.09 | 12.88 | 19.31 | 26.05 | 23.33 |

Being able to extract useful information from potentially noisy metadata is a desirable quality for a Disciple. Table 6 shows that using predicted metadata leads to only a decrease of ∼1-2% for MOGIC (OAK) across metrics showing that MOGIC (OAK) is fairly robust. Whereas, Oracle accuracy drops by 17% in P@1 when it uses predicted vs ground truth metadata. Thus, we leveraged ground truth metadata while training the Oracle model. Although MOGIC uses a less robust Oracle, the mechanisms in our Oracle-guidance based training helps train a significantly robust Disciple.

**MOGIC is generally applicable to any XC Disciple**: So far, we have presented results using OAK as the Disciple model. To test the general applicability of the MOGIC framework, we experiment with two other popular XC Disciples: NGAME and DEXA. Table 7 shows that MOGIC provides 1-2% improvement in precision and NDCG, and 2-3% improvement in PSP over the base XC algorithms.

Table 7: MOGIC is generally applicable to any XC Disciple: On LF-WikiSeeAlsoTitles-320K, MOGIC improves accuracy of the base algorithm by 1-2% in P@1. For results on LF-AmazonTitles-131K dataset refer Appendix E.

| Models | P@1 | P@5 | N@5 | PSP@1 | PSP@5 |
|---|---|---|---|---|---|
| MOGIC (OAK) | **34.62** | **17.93** | **27.44** | **35.70** | **33.18** |
| OAK | 33.71 | 17.12 | 24.53 | 33.83 | 30.83 |
| MOGIC (NGAME) | **32.37** | **16.38** | **26.87** | **33.16** | **31.08** |
| NGAME | 30.72 | 15.42 | 25.18 | 31.56 | 28.88 |
| MOGIC (DEXA) | **32.75** | **16.92** | **26.88** | **34.00** | **31.82** |
| DEXA | 31.57 | 16.14 | 25.64 | 32.71 | 29.99 |

## 4.3 ROBUSTNESS ANALYSIS

We perform two kinds of robustness analysis to test MOGIC: missing metadata and noisy metadata.

**Robustness to Missing Metadata**: MOGIC uses memory items to improve query representation and regularize XC models. In Table 8, we show results by reducing the size of the memory bank (*i.e.* randomly subsampling the set of retrieved memory items). This table shows that as the size of memory bank is decreased by randomly removing items, MOGIC's performance decreases only slightly (both with predicted as well as with ground truth metadata) but Oracle suffers significantly.

Notice that drop in accuracy for Oracle model is more as compared to MOGIC, which is because MOGIC is more robust to noise in choice of memory items due to our training procedure.

Table 8: Results by reducing the size of the memory bank on LF-WikiSeeAlsoTitles-320K. As the size of memory bank is decreased by randomly removing items, MOGIC's performance decreases only slightly (both with predicted as well as with ground truth metadata) but Oracle suffers significantly.

| | MOGIC + predicted metadata | | | | | MOGIC + groundtruth metadata | | | | | Oracle | | | | |
|---|---|---|---|---|---|---|---|---|---|---|---|---|---|---|---|
| Size % | P@1 | P@5 | N@5 | PSP@1 | PSP@5 | P@1 | P@5 | N@5 | PSP@1 | PSP@5 | P@1 | P@5 | N@5 | PSP@1 | PSP@5 |
| 100 | **34.62** | **17.93** | **27.44** | **35.70** | **33.18** | **36.94** | **19.12** | **29.00** | **38.42** | **35.07** | 42.78 | 20.53 | 32.99 | 43.59 | 37.57 |
| 80 | 34.54 | 17.87 | 27.36 | 35.60 | 33.08 | 36.69 | 19.02 | 28.82 | 38.17 | 34.92 | 39.22 | 19.07 | 30.44 | 40.15 | 35.08 |
| 60 | 34.38 | 17.81 | 27.29 | 35.47 | 32.98 | 36.47 | 18.86 | 28.70 | 37.85 | 34.66 | 35.08 | 17.30 | 27.58 | 36.13 | 32.08 |
| 40 | 34.17 | 17.72 | 27.22 | 35.28 | 32.85 | 35.91 | 18.59 | 28.45 | 37.24 | 34.26 | 30.64 | 15.43 | 24.45 | 31.80 | 28.92 |

**Robustness to Noisy Metadata**: To further understand the relationship between Oracle's performance and MOGIC' performance, we use ground truth metadata while predicting from both the Oracle model and the MOGIC model. Further, for every query, we inject different levels of noise (varying from 0% to 60%) to ground truth metadata, and check the degree of robustness of both the models to such noise. Noise is added by randomly replacing a certain percentage of ground truth metadata items by irrelevant ones. Table 9 shows that as we increase noise, XC task performance decreases for both the models signifying the importance of clean metadata. MOGIC's downstream performance decreases slightly while Oracle's performance decreases significantly showing the robustness of MOGIC.

Table 9: MOGIC is more robust to noisy metadata. Introducing noise in fused metadata at inference can lead to up to 20% reduction in accuracy of the Oracle, since early-fusion models rely on high-quality metadata at the input unlike the late-fusion-based MOGIC model and OAK models.

| | MOGIC (OAK) | | | | | Oracle | | | | | OAK | | | | |
|---|---|---|---|---|---|---|---|---|---|---|---|---|---|---|---|
| Noise % | P@1 | P@5 | N@5 | PSP@1 | PSP@5 | P@1 | P@5 | N@5 | PSP@1 | PSP@5 | P@1 | P@5 | N@5 | PSP@1 | PSP@5 |
| 0 | **36.94** | **19.12** | **29.00** | **38.42** | **35.07** | 42.78 | 20.53 | 32.99 | 43.59 | 37.57 | 35.28 | 17.97 | 36.35 | 28.23 | 33.13 |
| 20 | 36.26 | 18.8 | 28.66 | 37.69 | 34.61 | 34.8 | 16.83 | 26.67 | 35.64 | 30.73 | 34.64 | 17.63 | 35.61 | 27.9 | 32.66 |
| 40 | 35.62 | 18.44 | 28.36 | 36.9 | 34.08 | 26.75 | 13.1 | 20.45 | 27.56 | 23.87 | 34.17 | 17.33 | 34.99 | 27.66 | 32.23 |
| 60 | 34.92 | 18.12 | 27.94 | 36.19 | 33.59 | 18.65 | 9.31 | 14.29 | 19.44 | 17.02 | 33.54 | 17.02 | 34.31 | 27.33 | 31.79 |

## 5 CONCLUSION

We introduce MOGIC, a novel framework for enriching query representations using relevant metadata without incurring high inference latency. This is achieved via a two phase training. The first phase trains an Oracle using text metadata infusion both on the query as well as the label side. The second phase involves guiding the training of a Disciple model using embeddings from the Oracle classifier.

Through extensive experiments on four popular benchmark XC datasets, we have demonstrated that MOGIC significantly outperforms state-of-the-art XC models, achieving improvements in terms of precision, NDCG, and propensity scored precision. Moreover, MOGIC exhibits robustness to missing and noisy metadata, making it a valuable tool for real-world applications.

In conclusion, MOGIC represents a significant advancement in the field of extreme classification, offering a practical and effective solution for incorporating metadata to enhance model performance. Our work highlights the potential of Oracle-guided training for improving the robustness and accuracy of memory-based models in challenging classification tasks.

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

# Appendix

## Table of Contents

## A  THEORETICAL PROOFS

### A.1  NOTATIONS

Let $\left\{ \{(X_i, \mathbf{y}_i)\}_{i=1}^N, \{Z_l\}_{l=1}^L \right\}$ be the training dataset, where $X_i, Z_l$ are the raw text features of $i$th query and $l$th label respectively, and $\mathbf{y}_i \in \{0, 1\}^L$ is the binary label vector for the $i$th query.

Let the Oracle be a dual encoder model denoted by parameters $\theta_O = \{\theta_O^q, \theta_O^l\}$. Similarly, let the Disciple also be a dual encoder model denoted by parameters $\theta_D = \{\theta_D^q, \theta_D^l\}$. Given the raw text query and label samples $X$ and $Z$, let the frozen Oracle embeddings be denoted as $\mathbf{x}^* = \mathcal{E}(X|\theta_O), \mathbf{z}^* = \mathcal{E}(Z|\theta_O)$, and the trainable Disciple embeddings be denoted by $\mathbf{x} = \mathcal{E}(X|\theta_D), \mathbf{z} = \mathcal{E}(Z|\theta_D)$.

Let $\mathbf{Z} = \{\mathbf{z}_1, \cdots, \mathbf{z}_L\}$ be the matrix of all label embeddings stacked together. Consider the loss $\mathcal{L}(\{(\mathbf{x}_i, \mathbf{y}_i)\}_{i=1}^N, \{\mathbf{z}_l\}_{l=1}^L) = \frac{1}{N} \sum_{i=1}^N \ell(\mathbf{Z}^\top \mathbf{x}_i, \mathbf{y}_i)$ to be a generic loss function that is separable over query samples. Note that the triplet loss used by MOGIC falls in this family of loss functions, and therefore the analysis presented below holds true for it.

Let the query and label towers of the Disciple model $\theta_D = \{\theta_D^q, \theta_D^l\}$ belong to hypothesis classes $\mathcal{F}, \mathcal{G}$ whose complexities be bounded by Rademacher constants $R_q, R_l$ respectively.

Under the above setting, the alignment and matching losses can be expressed as:

$$\mathcal{L}_{\text{Alignment}} = \frac{1}{2}\Big(\mathcal{L}(\{(\mathbf{x}_i, \mathbf{y}_i)\}_{i=1}^N, \{\mathbf{z}_l^*\}_{l=1}^L) + \mathcal{L}(\{(\mathbf{x}_i^*, \mathbf{y}_i)\}_{i=1}^N, \{\mathbf{z}_l\}_{l=1}^L)\Big) \tag{4}$$

$$\mathcal{L}_{\text{Matching}} = \frac{1}{N}\sum_{i=1}^N \|\mathbf{x}_i - \mathbf{x}_i^*\| + \frac{1}{L}\sum_{l=1}^L \|\mathbf{z}_l - \mathbf{z}_l^*\| \tag{5}$$

## A.2 DERIVATIONS

The below lemma shows that the triplet loss used in MOGIC is upper-bounded by a linear combination of the Alignment and Matching loss, assuming that the individual pairwise loss terms comprising the triplet loss are Lipschitz continuous by themselves.

**Lemma 2.** *Let $P$ be the number of positive labels for a given query and, $\ell(\mathbf{s}_i, \mathbf{y}_i) = \frac{1}{P.(L-P)}\sum_{p,q\in\{1,\cdots,L\}}\mathbb{1}(y_{ip} = +1)\mathbb{1}(y_{in} = -1)g(s_{in} - s_{ip})$ be the triplet loss for a query $i$ which is decomposable over all relevant-irrelevant label pairs. If $g$ is Lipschitz-continuous with constant $K$, then the following inequalities hold true:*

$$\mathcal{L}(\{(\mathbf{x}_i, \mathbf{y}_i)\}_{i=1}^N, \{\mathbf{z}_l\}_{l=1}^L) \leq \mathcal{L}(\{(\mathbf{x}_i, \mathbf{y}_i)\}_{i=1}^N, \{\mathbf{z}_l^*\}_{l=1}^L) + \frac{2.K.B}{L}\sum_{l=1}^L \|\mathbf{z}_l - \mathbf{z}_l^*\|_2 \tag{6}$$

$$\mathcal{L}(\{(\mathbf{x}_i, \mathbf{y}_i)\}_{i=1}^N, \{\mathbf{z}_l\}_{l=1}^L) \leq \mathcal{L}(\{(\mathbf{x}_i^*, \mathbf{y}_i)\}_{i=1}^N, \{\mathbf{z}_l\}_{l=1}^L) + \frac{2.K.B}{N}\sum_{i=1}^N \|\mathbf{x}_i - \mathbf{x}_i^*\|_2 \tag{7}$$

*Proof.* Let $\mathbf{s}_i, \mathbf{s}_i'$ be two score vectors. Then,

$$|\ell(\mathbf{s}_i, \mathbf{y}_i) - \ell(\mathbf{s}_i', \mathbf{y}_i)| \tag{8}$$

$$= |\frac{1}{P.(L-P)}\sum_{p,q\in\{1,\cdots,L\}}\mathbb{1}(y_{ip} = +1)\mathbb{1}(y_{in} = -1)(g(s_{in} - s_{ip}) - g(s_{in}' - s_{ip}'))| \tag{9}$$

$$\leq \frac{1}{P.(L-P)}\sum_{p,q\in\{1,\cdots,L\}}\mathbb{1}(y_{ip} = +1)\mathbb{1}(y_{in} = -1)|g(s_{in} - s_{ip}) - g(s_{in}' - s_{ip}')| \tag{10}$$

$$\leq \frac{k}{P.(L-P)}\sum_{p,q\in\{1,\cdots,L\}}\mathbb{1}(y_{ip} = +1)\mathbb{1}(y_{in} = -1)(|s_{in} - s_{in}'| + |s_{ip} - s_{ip}'|) \tag{11}$$

If $\mathbf{s}_i = \mathbf{Z}^\top\mathbf{x}_i$ and $\mathbf{s}_i' = \mathbf{Z}^{*\top}\mathbf{x}_i$, then:

$$\frac{1}{N}\sum_{i=1}^N |\ell(\mathbf{s}_i, \mathbf{y}_i) - \ell(\mathbf{s}_i', \mathbf{y}_i)| \tag{12}$$

$$\leq \frac{K.B}{N.P.(L-P)}\sum_{p,q\in\{1,\cdots,L\}}\mathbb{1}(y_{ip} = +1)\mathbb{1}(y_{in} = -1)(\|\mathbf{z}_n - \mathbf{z}_n^*\|_2 + \|\mathbf{z}_p - \mathbf{z}_p^*\|_2) \tag{13}$$

$$\leq \frac{K.B}{N.P.(L-P)}\frac{2NP(L-P)}{L}\sum_{l=1}^L \|\mathbf{z}_l - \mathbf{z}_l^*\|_2 \tag{14}$$

$$= \frac{2.K.B}{L}\sum_{l=1}^L \|\mathbf{z}_l - \mathbf{z}_l^*\|_2 \tag{15}$$

As a result, the following inequality holds true:

$$\mathcal{L}(\{(\mathbf{x}_i, \mathbf{y}_i)\}_{i=1}^N, \{\mathbf{z}_l\}_{l=1}^L) \leq \mathcal{L}(\{(\mathbf{x}_i, \mathbf{y}_i)\}_{i=1}^N, \{\mathbf{z}_l^*\}_{l=1}^L) + \frac{2.k.B}{L} \sum_{l=1}^L \|\mathbf{z}_l - \mathbf{z}_l^*\|_2 \quad (16)$$

Similarly, if $\mathbf{s}_i = \mathbf{Z}^\top \mathbf{x}_i$ and $\mathbf{s}_i' = \mathbf{Z}^\top \mathbf{x}_i^*$, then:

$$\frac{1}{N} \sum_{i=1}^N |\ell(\mathbf{s}_i, \mathbf{y}_i) - \ell(\mathbf{s}_i', \mathbf{y}_i)| \quad (17)$$

$$\leq \frac{2.K.B}{N.P.(L-P)} \sum_{p,q \in \{1, \cdots, L\}} \mathbb{1}(y_{ip} = +1) \mathbb{1}(y_{in} = -1)(\|\mathbf{x}_i - \mathbf{x}_i^*\|_2) \quad (18)$$

$$= \frac{2.K.B}{N} \sum_{i=1}^N \|\mathbf{x}_i - \mathbf{x}_i^*\|_2 \quad (19)$$

As a result, the following inequality holds true as well:

$$\mathcal{L}(\{(\mathbf{x}_i, \mathbf{y}_i)\}_{i=1}^N, \{\mathbf{z}_l\}_{l=1}^L) \leq \mathcal{L}(\{(\mathbf{x}_i^*, \mathbf{y}_i)\}_{i=1}^N, \{\mathbf{z}_l\}_{l=1}^L) + \frac{2.k.B}{N} \sum_{i=1}^N \|\mathbf{x}_i - \mathbf{x}_i^*\|_2 \quad (20)$$

$\square$

**Lemma 3.** *Assume a realizable setting where, for some $\theta_D = \theta_D^*$, $\mathbf{x} = \mathbf{x}^*, \mathbf{z} = \mathbf{z}^*$ holds for all $\mathbf{x}, \mathbf{z}$. Now, let $\theta_D = \bar{\theta}_D$ be another value of $\theta_D$ which minimizes the Oracle-guided population loss $\mathcal{L}_{\text{Alignment}} + \lambda \mathcal{L}_{\text{Matching}}$. Its corresponding embeddings are denoted by $\bar{\mathbf{x}}, \bar{\mathbf{z}}$. Further, assume that all the embeddings are bounded by $\|\bar{\mathbf{x}}\|_2, \|\bar{\mathbf{z}}\|_2, \|\mathbf{x}^*\|_2, \|\mathbf{z}^*\|_2 \leq B$. Then, for $\lambda = KB$, the following inequalities hold:*

$$\mathcal{L}(\{(\bar{\mathbf{x}}_i, \mathbf{y}_i)\}_{i=1}^N, \{\bar{\mathbf{z}}_l\}_{l=1}^L) \leq \mathcal{L}_{\text{Alignment}} + \lambda \mathcal{L}_{\text{Matching}} \leq \mathcal{L}(\{(\mathbf{x}_i^*, \mathbf{y}_i)\}_{i=1}^N, \{\mathbf{z}_l^*\}_{l=1}^L) \quad (21)$$

*Proof.* By averaging the two inequalities in Lemma 2, we get the following result:

$$\mathcal{L}(\{(\bar{\mathbf{x}}_i, \mathbf{y}_i)\}_{i=1}^N, \{\bar{\mathbf{z}}_l\}_{l=1}^L) \quad (22)$$

$$\leq \frac{1}{2}(\mathcal{L}(\{(\bar{\mathbf{x}}_i, \mathbf{y}_i)\}_{i=1}^N, \{\mathbf{z}_l^*\}_{l=1}^L) + \mathcal{L}(\{(\mathbf{x}_i^*, \mathbf{y}_i)\}_{i=1}^N, \{\bar{\mathbf{z}}_l\}_{l=1}^L)) \quad (23)$$

$$+ K.B(\frac{1}{L} \sum_{l=1}^L \|\bar{\mathbf{z}}_l - \mathbf{z}_l^*\|_2 + \frac{1}{N} \sum_{i=1}^N \|\bar{\mathbf{x}}_i - \mathbf{x}_i^*\|_2) \quad (24)$$

$$= \mathcal{L}_{\text{Alignment}} + K.B.\mathcal{L}_{\text{Matching}} \quad (25)$$

Next, note that $\bar{\mathbf{x}}, \bar{\mathbf{z}}$ are the values of $\mathbf{x}, \mathbf{z}$ which minimize the Oracle-guided loss. Due to this and the realizable setting assumption:

$$\mathcal{L}_{\text{Alignment}} + K.B.\mathcal{L}_{\text{Matching}} \quad (26)$$

$$= \frac{1}{2}(\mathcal{L}(\{(\bar{\mathbf{x}}_i, \mathbf{y}_i)\}_{i=1}^N, \{\mathbf{z}_l^*\}_{l=1}^L) + \mathcal{L}(\{(\mathbf{x}_i^*, \mathbf{y}_i)\}_{i=1}^N, \{\bar{\mathbf{z}}_l\}_{l=1}^L)) \quad (27)$$

$$+ K.B(\frac{1}{L} \sum_{l=1}^L \|\bar{\mathbf{z}}_l - \mathbf{z}_l^*\|_2 + \frac{1}{N} \sum_{i=1}^N \|\bar{\mathbf{x}}_i - \mathbf{x}_i^*\|_2) \quad (28)$$

$$\leq \frac{1}{2}(\mathcal{L}(\{(\mathbf{x}_i^*, \mathbf{y}_i)\}_{i=1}^N, \{\mathbf{z}_l^*\}_{l=1}^L) + \mathcal{L}(\{(\mathbf{x}_i^*, \mathbf{y}_i)\}_{i=1}^N, \{\mathbf{z}_l^*\}_{l=1}^L)) \quad (29)$$

$$+ K.B(\frac{1}{L} \sum_{l=1}^L \|\mathbf{z}_l^* - \mathbf{z}_l^*\|_2 + \frac{1}{N} \sum_{i=1}^N \|\mathbf{x}_i^* - \mathbf{x}_i^*\|_2) \quad (30)$$

$$= \mathcal{L}(\{(\mathbf{x}_i^*, \mathbf{y}_i)\}_{i=1}^N, \{\mathbf{z}_l^*\}_{l=1}^L) \quad (31)$$

The above proves the two inequalities. $\square$

However, optimizing the population-level Oracle-guided loss is not feasible as we are often restricted to a finite training sample size. Now, empirical loss optimization on the finite training set introduces some error. The following lemma bounds this error:

**Lemma 4.** *Let the Disciple model be trained by minimizing the Oracle-guided loss on the training set $\mathcal{D} = \left\{ \{(X_i, \mathbf{y}_i)\}_{i=1}^N, \{Z_l\}_{l=1}^L \right\}$. Let the empirical training risk attained by this minimization be $\hat{\mathcal{L}}$, then the following inequality holds:*

$$|\min_{\theta_D} \mathbb{E}\mathcal{L} - \hat{\mathcal{L}}| \leq \frac{2K}{N}.(R_q + R_l) + \sqrt{\frac{\log(\frac{1}{\delta})}{N}} \tag{32}$$

*where $\mathcal{L}$ is the population-level Oracle-guided training loss.*

*Proof.* Proof uses the standard ideas of ghost sampling and Rademacher complexity bounding, along with some well-known properties of Rademacher complexity. Note here that $\min_{\theta_D} \mathbb{E}\mathcal{L} = \mathcal{L}_{\texttt{Alignment}} + \lambda \mathcal{L}_{\texttt{Matching}}$ with $\bar{\mathbf{x}}, \bar{\mathbf{z}}$ embeddings, thus connecting to Lemma 3. $\qquad \square$

**Theorem 5.** *Given the problem setting described above, if the Disciple model is trained by minimizing the Oracle-guided loss $\mathcal{L} = \mathcal{L}_{\texttt{Alignment}} + \lambda \mathcal{L}_{\texttt{Matching}}$ on the training set $\mathcal{D} = \left\{ \{(X_i, \mathbf{y}_i)\}_{i=1}^N, \{Z_l\}_{l=1}^L \right\}$, then for some $\lambda > 0$ and any $\delta \in [0, 1]$, the following inequality holds true with prob. at least $1 - \delta$:*

$$\mathbb{E}_{(\mathbf{x},\mathbf{y})}\ell(\mathbf{Z}^\top\mathbf{x}, \mathbf{y}) \leq \mathbb{E}_{(\mathbf{x},\mathbf{y})}\ell(\mathbf{Z}^{*\top}\mathbf{x}^*, \mathbf{y}) + \frac{4K}{N}.(R_q + R_l) + 2\sqrt{\frac{\log(\frac{1}{\delta})}{N}} \tag{33}$$

*Proof.* Proof involves a simple algebraic combination of the results in Lemmas 3 and 4. $\qquad \square$

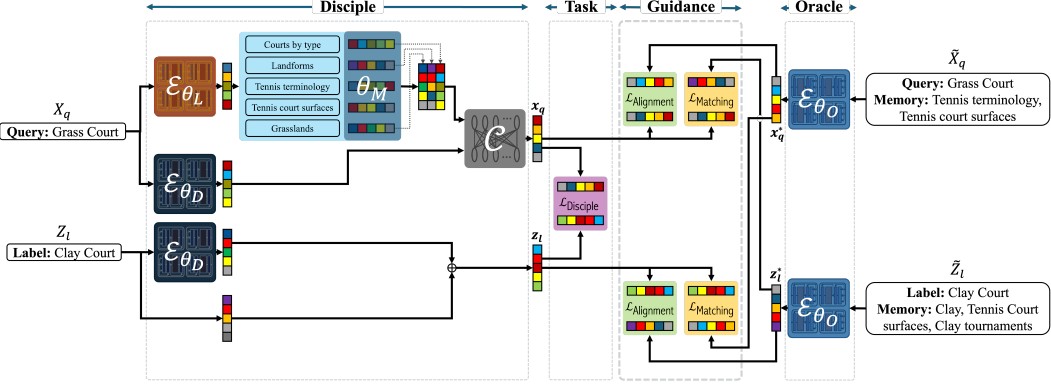

Figure 3: This is a detailed version of Figure 1. $\theta_L$ represents the parameters of the classifier that predicts metadata for a given query. $\theta_M$ denotes the memory bank containing memory items corresponding to query and label metadata. $\theta_D$ represents the parameters of the base encoder in Disciple. $\mathcal{C}$ signifies the combined module. Collectively, $\theta_L$, $\theta_M$, $\theta_E$, and $\mathcal{C}$ constitute the parameters of the Disciple, denoted as $\theta_{\mathcal{D}}$.

## B  END-TO-END WALKTHROUGH

This section provides a detailed walkthrough of the MOGIC framework as depicted in Figure 1. The Figure 3 shows a more detailed version of the architecture and it will help us to walk through an example on the training and the inference of the MOGIC framework. Let us consider the query "Grass Court" mentioned in Table 1. The framework has three major blocks for Query Processing block, Label Processing block and Oracle representation block. The Query Processing block and Label Processing block are part of the Disciple.

**Query Processing block:** involves the following steps

- **Metadata Retrieval:** The query "Grass Court" is sent to the memory bank to retrieve relevant metadata, including "Courts by type", "Landforms", and "Grasslands". The memory bank contains vector representations for each of these metadata. These vector representations are then sent for further processing.
- **Query Encoding:** The query "Grass court" is passed through the main encoder to obtain its vector representation.
- **Query Enrichment:** The query representation is fused with the metadata representation using a cross-attention layer to create an enriched query representation.

**Label Processing block:** similar to the above query processing

- **Label Encoding:** The label "Clay Court" is passed through the main encoder to obtain its vector representation. This encoder is shared between query and the label.
- **Label Enrichment:** This representation is further enriched by combining it with a separate "Clay Court" free parameter.

**Oracle representation block:** The query "Grass Court" is concatenated with its associated metadata "Tennis terminology", "Sports rules and regulations", "Tennis court surfaces" to form a super query "Clay Court Tennis terminology Sports rules and regulations Tennis court surfaces". This super query is passed through the Oracle encoder to obtain its vector representation. Similarly "Clay court" is concatenated with "Clay", "Tennis court surfaces" and "Clay tournaments" and passed to the shared Oracle encoder to obtain its vector representation.

The above blocks are then used for training and inference,

**Inference:** During inference for the query "Grass Court" we compute its query representation using the Query processing block and then calculate its similarity with all label representations computed from the Label processing block in the dataset, including "Alabama", "Clay Court", "Carpet Court", "Hardcourt" and "Henry Moore", to determine their relevance to the query using cosine similarity distance.

**Training:** Unlike inference, training uses all the blocks and involves the following steps:

- **Vector Representations:** We compute query and label representations for both the query "Grass Court" and the label "Clay Court" using the Query processing block and the Label processing block.
- **Triplet Loss:** We then apply triplet loss to the query and label representations, as is common in retrieval methods.
- **Oracle Regularization:** To further regularize the Disciple model, we introduce additional triplet loss terms (a) between the query "Grass Court" Oracle representation and the "Clay Court" label representation. (b) between the query "Clay Court" Oracle representation and the "Grass Court" label representation. (c) mean squared error (MSE) loss to minimize the distance between the "Grass Court" Oracle representation and its query representation and the "Clay Court" Oracle representation and its label representation.

**What makes MOGIC (OAK) perform better than OAK?**: We observe that for the first example, "Courts by type" is a good memory item, but some predicted memory items ("Landforms" and

"Grasslands") are misleading. This is the same metadata which is used both by OAK as well as MOGIC (OAK). Unfortunately, the misleading metadata causes OAK to produce bad predictions about geographical places like "Texas, List of Nevada state prisons, Ronald Reagan Boyhood Home, West End (Richmond, Virginia)". The Oracle-guidance fortunately helped MOGIC (OAK) to avoid paying attention to the misleading metadata and therefore MOGIC (OAK) ends up predicting accurate labels like "Clay court, Carpet court, Hardcourt". Even the label "U.S. Men's Clay Court Championships" is somewhat relevant. With MOGIC regularized training, MOGIC (OAK) was able to retain the original intent of the query and predicted various type tennis courts.

Table 10: The synergistic combination of the various components within MOGIC leads to the observed performance gains. This table shows ablations of MOGIC with different components on LF-WikiSeeAlsoTitles-320K dataset.

| Setting | Models | P@1 | P@5 | N@5 | PSP@1 | PSP@5 |
|---------|--------|-----|-----|-----|-------|-------|
| 1 | MOGIC (OAK) | **34.62** | **17.93** | **35.70** | **27.44** | **33.18** |
| 2 | MOGIC (OAK) + Oracle w/o Metadata | 34.09 | 17.43 | 34.90 | 26.95 | 32.13 |
| 3 | MOGIC (OAK) on 'OAK + ground-truth Metadata' | 34.25 | 17.75 | 35.45 | 26.81 | 26.81 |
| 4 | Early Fusion (similar to REALM) | 28.49 | 14.52 | 29.46 | 22.26 | 26.52 |
| 5 | MOGIC (OAK) + Early Fusion | 29.30 | 14.88 | 29.92 | 22.21 | 26.87 |

## C    CONTRIBUTION OF COMPONENTS TO PERFORMANCE GAIN

To systematically evaluate the individual contributions of the various components within the MOGIC framework, we conduct a series of ablation studies in this section (cf. Table 10).

**Contribution of metadata in Oracle training (Setting 2):** We train the MOGIC (OAK) model using an Oracle which has not been trained using any metadata. As seen in Table 10, this experiment yields results that are better than standard OAK but not as good as the proposed MOGIC (OAK) where the Oracle has access to metadata during training. This shows that while individually, the regularization itself can result in better performance of the Disciple, training Oracle with access to metadata further improves the performance.

**Training the Disciple OAK with ground truth metadata in MOGIC (OAK) (Setting 3):** Broadly, MOGIC performs training of the Disciple model with the predicted metadata as opposed to OAK (Mohan et al., 2024) which uses ground-truth metadata. To validate this choice, we perform an ablation where the Disciple is trained with the ground-truth metadata. Table 10 setting 3 shows that using ground-truth metadata for the Disciple performs worse than our proposed framework. This is due to the potential mismatch between training and inference distributions when the Disciple is trained solely with ground-truth metadata. While the combiner can filter out unnecessary metadata, maintaining a closer training-inference distribution proves helpful. This experiment shows that it would not be beneficial to train the models directly with ground truth metadata.

**Early fusion in Disciple model (Setting 4 and 5):** We choose to perform a late fusion of the metadata information in the Disciple model by passing the query representation and memory items through a single layer combiner. An alternate choice could be to perform early fusion of metadata. To validate this, we perform two experiments, (a) only early fusion in Disciple (Setting 4), where the predicted metadata is concatenated with the input and (b) both early and late fusion (Setting 5) where alongside concatenating the predicted metadata at input (Table 10), we also perform the fusion using the combiner layer. Both of these approaches perform significantly worse than our proposed framework. This is because late fusion adds robustness to incorrect predictions in the metadata.

## D    EFFICIENCY ANALYSIS

We now report the training and inference time and comparisons between Oracle and MOGIC (OAK). The Table 11 and Table 12 summarizes these numbers. As we already note in Section 4, MOGIC is merely a regularization framework, and we observe that inference times for a given baseline Disciple

model and its MOGIC variant are identical. There is marginal increase in the training time due to computation of the additional regularization terms in the loss function.

**Training**: All models were trained on AMD 4xMI200 GPUs. MOGIC (OAK) uses a context length of 32 for short-text datasets and 256 for full-text datasets. The Oracle is trained with context length of 128 and 256 for short and full-text datasets, respectively. The training time for DistilBERT Oracle is roughly 2x that of the MOGIC (OAK) due to the increased context length for short-text datasets. We attribute the higher training time for the Oracle (for short text datasets) to its longer context length leading to larger attention matrices used to calculate backward-pass gradients.

**Inference**: We provide inference latency of MOGIC (OAK) for different datasets on CPU, 8 threads, single query, 2 metadata vectors considering a PyTorch implementation. For short text datasets, MOGIC (OAK) is much faster. For full text datasets, since the query size is large by itself, the overall size of early concatenated text input for the Oracle is similar to the size of the input for the Disciple. Hence, inference times are almost similar for Disciple and Oracle for full-text datasets.

We also provide end-to-end production latency to validate real-world feasibility, with inference on CPU, with single query, and two metadata vectors. Model optimizations allow for much smaller latencies, reducing the gap between OAK encoder and MOGIC (OAK) inference latency - making the latter feasible.

Table 11: Training and inference time of MOGIC (OAK) and DistilBERT Oracle on different datasets.

| | MOGIC (OAK) | | DistilBERT Oracle | |
|---|---|---|---|---|
| Dataset | Inference (in ms) | Training (in hrs) | Inference (in ms) | Training (in hrs) |
| LF-WikiSeeAlsoTitles-320K | 14.06 | 25 | 24.70 | 45 |
| LF-WikiTitles-500K | 13.72 | 41 | 26.79 | 69 |
| LF-WikiSeeAlso-320K | 52.85 | 88 | 49.56 | 90 |
| LF-Wikipedia-500K | 50.27 | 180 | 48.88 | 173 |
| LF-AmazonTitles-131K | 13.66 | 8 | 25.21 | 13 |
| LF-Amazon-131K | 51.91 | 22 | 49.99 | 23 |

Table 12: Real world deployment latency of MOGIC (OAK) and its OAK encoder in milliseconds.

| Algorithm | Mean | p95 | p99 | p999 |
|---|---|---|---|---|
| OAK encoder | 10.30 | 20.05 | 22.59 | 23.49 |
| MOGIC (OAK) | 14.95 | 22.34 | 25.07 | 27.08 |

# E    GENERALIZATION OF MOGIC ON OTHER DATASETS ACROSS DISCIPLES

MOGIC can generalize across both Disciples and datasets. To validate this, we now include results on training MOGIC with the DEXA and NGAME Disciples on LF-Amazon-131K dataset. Table 13 summarizes these results, wherein we observe that MOGIC demonstrates performance gains across both Disciples.

Table 13: MOGIC is a general framework, and can be extended to any base XC algorithm to improve its accuracy. Along with LF-WikiSeeAlsoTitles-320K (Table 7), MOGIC also shows gains over other Disciples on LF-AmazonTitles-131K. We observe MOGIC can improve accuracy of the base algorithm by 1-2% in P@1.

| Models | P@1 | P@5 | N@5 | PSP@1 | PSP@5 |
|---|---|---|---|---|---|
| MOGIC (OAK) | 47.01 | 22.40 | 49.51 | 40.62 | 50.33 |
| OAK | 46.42 | 21.88 | 49.06 | 39.76 | 49.78 |
| MOGIC (NGAME) | 44.27 | 21.26 | 47.48 | 39.48 | 49.18 |
| NGAME | 43.44 | 21.16 | 47.10 | 39.00 | 49.00 |
| MOGIC (DEXA) | 45.43 | 21.70 | 48.49 | 39.91 | 49.95 |
| DEXA | 44.47 | 21.34 | 47.65 | 39.25 | 49.08 |

Table 15: Hyper-parameter values for MOGIC on all datasets to enable reproducibility. MOGIC code will be released publicly. Most hyperparameters were set to their default values across all datasets. **LR** is learning rate. Margin $\gamma = 0.3$ was used for contrastive loss. A cell containing the symbol $\uparrow$ indicates that that cell contains the same hyperparameter value present in the cell directly above it.

| Dataset | Batch Size $S$ | Encoder epochs | Encoder LR $lr$ | Bert seq. length $L_{max}$ |
|---|---|---|---|---|
| LF-WikiSeeAlsoTitles-320K | 1024 | 300 | 0.0002 | 32 |
| LF-WikiTitles-500K | $\uparrow$ | $\uparrow$ | $\uparrow$ | $\uparrow$ |
| LF-WikiSeeAlso-320K | $\uparrow$ | $\uparrow$ | $\uparrow$ | 256 |
| LF-Wikipedia-500K | $\uparrow$ | $\uparrow$ | $\uparrow$ | $\uparrow$ |

# F  ORACLE MODELS WITH LINEAR PROJECTION LAYER

Using LLaMA-2 or Phi-2 as an Oracle mandates the use of a down-projection layer, as the models operate with a 4096-dimensional space, while Disciple models such as OAK operate in a 768-dimensional space. As an alternative to the PCA model, we now also consider a learnable projection transform atop the Oracle. We observe that (cf. Table 14) the 7B LoRA-fine-tuned LLaMA-2 Oracle with a projection layer performs on par with the 66M parameter standard-fine-tuned DistilBERT Oracle. This presents a performance versus training overhead trade-off between using small, but task-specific, Oracles and large pre-trained, but fine-tuned, general purpose Oracles, which is a promising direction for future research.

Table 14: MOGIC on the LF-WikiSeeAlsoTitles-320K dataset with different Oracle models and an addition projection layer

| Oracle Models | Finetuning | MOGIC (OAK) | | | | |
|---|---|---|---|---|---|---|
| | | P@1 | P@5 | N@5 | PSP@1 | PSP@5 |
| DistilBERT (65M params) | full | 34.62 | 17.93 | 35.70 | **27.44** | **33.18** |
| LLaMA-2 (7B params) | LoRA | **34.64** | **17.93** | **35.71** | 27.28 | 33.02 |
| Phi-2 (2.7B params) | LoRA | 34.34 | 17.73 | 35.44 | 27.09 | 32.71 |

# G  HYPER PARAMETERS AND TRAINING DETAILS

Table 15 shows hyper-parameters used in MOGIC to regularize XC models. SLMs were obtained from the HuggingFace model repository. Phi-2 (2.7B parameters) was retrieved from `https://huggingface.co/microsoft/phi-2`, and LLaMA-2 (7B parameters) was retrieved from `https://huggingface.co/meta-llama/Llama-2-7b`.

# H  PROMPT FOR LLMS

```
Given the title of a wikipedia article and the corresponding categories of that article on
wikipedia, your task is to predict the titles of all articles which are likely to be listed
in the see also section of the mentioned article. Output the coma separated list of titles of
the articles in the see also section of the given article.

\#\#\# Input : \newline
\#\#\# Title : agricultural science \newline
\#\#\# Categories : agriculture, agronomy \newline

\#\#\#\# Task Output \newline
\#\#\#\# Predicted titles \newline
agricultural sciences basic topics, agriculture ministry, agroecology, american society of
agronomy, genomics of domestication, history of agricultural science, institute of food and
agricultural sciences, international assessment of agricultural science and technology for
development, national ffa organization, agricultural science.
```

Listing 1: Prompt used for LoRA-finetuning small language models (LLaMA-2 and Phi-2)

## I    EVALUATION METRICS

Performance has been evaluated using propensity scored precision@$k$ and nDCG@$k$, which are unbiased and more suitable metric in the extreme multi-labels setting (Jain et al., 2016; Babbar & Schölkopf, 2019; Prabhu et al., 2018a;b). The propensity model and values available on The Extreme Classification Repository (Bhatia et al., 2016) were used. Performance has also been evaluated using vanilla precision@$k$ and nDCG@$k$ (with $k = 1, 3$ and $5$) for extreme classification.

Let $\hat{\mathbf{y}} \in \mathbb{R}^L$ denote the predicted score vector and $\mathbf{y} \in \{0,1\}^L$ denote the ground truth vector (with $\{0,1\}$ entries this time instead of $\pm 1$ entries, for sake of convenience). The notation $rank_k(\hat{\mathbf{y}}) \subset [L]$ denotes the set of $k$ labels with highest scores in the prediction score vector $\hat{\mathbf{y}}$ and $\|\mathbf{y}\|_1$ denotes the number of relevant labels in the ground truth vector. Then we have:

$$P@k = \frac{1}{k} \sum_{l \in rank_k(\hat{\mathbf{y}})} y_l$$

$$PSP@k = \frac{1}{k} \sum_{l \in rank_k(\hat{\mathbf{y}})} \frac{y_l}{p_l}$$

$$DCG@k = \frac{1}{k} \sum_{l \in rank_k(\hat{\mathbf{y}})} \frac{y_l}{\log(l+1)}$$

$$PSDCG@k = \frac{1}{k} \sum_{l \in rank_k(\hat{\mathbf{y}})} \frac{y_l}{p_l \log(l+1)}$$

$$nDCG@k = \frac{DCG@k}{\sum_{l=1}^{\min(k,\|\mathbf{y}\|_0)} \frac{1}{\log(l+1)}}$$

$$PSnDCG@k = \frac{PSDCG@k}{\sum_{l=1}^{k} \frac{1}{\log l+1}}$$

$$FN@k = 1 - \frac{\sum_{l \in rank_k(\hat{\mathbf{y}})} y_l}{\|\mathbf{y}\|_1}$$

Here, $p_l$ is propensity score of the label $l$ calculated as described in Jain et al. (2016).

## J    LABEL QUANTILE CREATION

For Figure 2 labels were divided into 5 equi-voluminous quantiles. To each label $l \in [L]$, a popularity score $V_l = |i : y_{il} = +2|$ was assigned by counting number of training datapoints tagged with that label. The total volume of all labels was computed as $V_{\text{tot}} \overset{\text{def}}{=} \sum_{l \in [L]} V_l$. Labels were arranged in decreasing order of their popularity score $V_l$. 5 label quantiles were then created so that the volume of labels in each bin is roughly $\approx V_{\text{tot}}/5$. Thus, labels were collected in the first bin in decreasing order of popularity till the total volume of labels in that bin exceeded $V_{\text{tot}}/5$ at which point the first bin was complete and the second bin was created by selecting remaining labels in decreasing order or popularity till the total volume of labels in the second bin exceeded $V_{\text{tot}}/5$ and so on. For example, for the LF-WikiTitles-500K dataset, the five bins were found to contain approximately $1K, 9K, 30K, 84K, 375K$ labels respectively. Note that the first bin contains very few $\approx 1K$ labels since these are head labels and a small number of them quickly racked up a total volume of $\approx V_{\text{tot}}/5$ whereas the last quantile contains more than $100\times$ more labels at around $375K$ labels since these are tail labels and so a lot more of them are needed to add up to a total volume of $\approx V_{\text{tot}}/5$.

## K    RESULTS ON AMAZON DATASETS

We observe that MOGIC achieves a P@1 of 50.05, which is a 1.68 gain over the 48.36 reported by the best baseline OAK on LF-Amazon-131K. MOGIC also obtains a P@1 of 47.01, which is a 0.6

gain over the 46.42 reported by the best baseline, OAK on LF-AmazonTitles-131K (cf. Table 16). This validates the strong generalizability of the MOGIC algorithm.

Table 16: Results on Amazon benchmark datasets. MOGIC is up to 2% more accurate as compared to baselines.

| Method | P@1 | P@5 | N@5 | PSP@1 | PSP@5 | P@1 | P@5 | N@5 | PSP@1 | PSP@5 |
|---|---|---|---|---|---|---|---|---|---|---|
| | LF-AmazonTitles-131K | | | | | LF-Amazon-131K | | | | |
| MOGIC (OAK) | **47.01** | **22.40** | **49.51** | **40.62** | **50.33** | **50.05** | **23.72** | **52.87** | **41.90** | **53.80** |
| OAK | *46.42* | *21.88* | *49.06* | *39.76* | *49.78* | *48.36* | *22.20* | *51.27* | *40.26* | *52.21* |
| DEXA | 46.42 | 21.59 | 49.00 | - | 49.65 | 46.64 | 22.06 | - | 38.83 | 50.38 |
| NGAME | 46.01 | 21.47 | 48.67 | 38.81 | 49.43 | 46.53 | 22.02 | 49.58 | 38.53 | 50.45 |

## L  PRELIMINARIES

- **Ground-truth metadata:** Besides the query text, often, a variety of auxiliary information is available in many domains, e.g., frequently clicked webpages for search queries in sponsored search, previously searched queries for web search query auto-completion, etc. Auxiliary information available from disparate but related tasks often have relevant diverse information that the input query does not, which can be leveraged to provide better predictions. We call such auxiliary information as ground-truth metadata. Metadata often has relevant diverse information that the input query does not, which can be leveraged to provide better predictions. For example, on sponsored search ads task that involves query-to-ad-keyword prediction, the query-side metadata is obtained by mining the organic search webpage titles clicked in response to the query on the search engine, while on the Wikipedia categories prediction task, other Wikipedia article titles connected to the original page via hyperlinks could serve as the metadata.

- **Early fusion**: When the query and metadata tokens are concatenated in the original text form itself (initial stage of processing) rather than in embedding form, we call it early fusion. This approach contrasts with late fusion, where these are combined at later stages.

- **Late-stage fusion**: When the query and metadata embeddings are combined after their tokens have been processed through multiple Transformer encoder layers, we call this combination as late-stage fusion.

- **Metadata-infused Oracle**: This is an Oracle model which is given access to ground truth metadata. Metadata-infused Oracle is the crux of our MOGIC framework. In the two stage method, Metadata-infused Oracle training forms the first stage. In this stage, an Oracle model is trained using both query-side and label-side ground-truth metadata. This metadata is in textual form and is used to enhance the training process.

- **Memory-based models versus Memory-free models**: Memory-based XC models are models that have access to memory (metadata). Very few XC methods are memory-based. On the other hand, most of the XC methods do not leverage metadata at all and are therefore called memory-free methods.

- **Query-side metadata**: Additional auxiliary information related to the input query is called query-side metadata. For example, in Table 1, for the query "Grass court", query-side metadata can be "Tennis terminology", "Sports rules and regulations", "Tennis court surfaces".

- **Label-side metadata**: Additional auxiliary information related to a label is called label-side metadata. For example, for the label "Clay court", label-side metadata is "Tennis terminology", "Sports rules and regulations", "Clay", "Tennis court surfaces".

- **Missing labels**: XC tasks involve a vast number of labels, making it impractical for annotators to mark all potential labels. This inherent limitation often results in missing labels, which are labels that should have been included in the ground truth but were inadvertently omitted.

- **Memory bank**: Memory bank is associated with both the queries and labels, having parameters $\theta_M$. Formally, memory bank is represented by $\mathcal{M} \in \mathbb{R}^{M \times D}$, each memory

item $j$ is mapped to a row in the matrix which we call its memory item representation $\mathbf{m}_j \in \mathbb{R}^D$. Here $\mathcal{M}(\cdot|\theta_M)$ returns relevant memory items for query $X_q$ and label $Z_l$, i.e., $\mathbf{a}_{q/l} \in \mathcal{M}(X_q/Z_l|\theta_M)$.

## M   VISUALIZATION OF MOGIC (OAK)

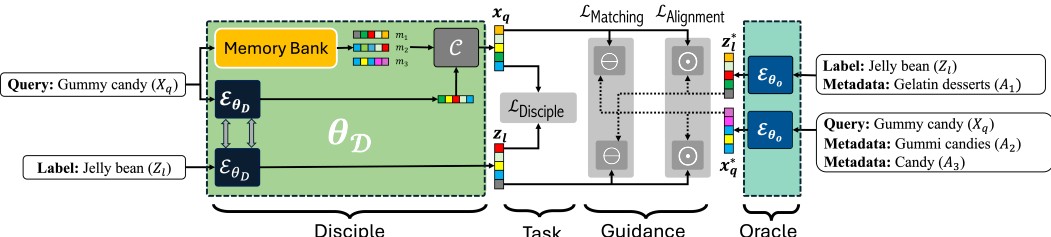

Figure 4: MOGIC (OAK) training framework, depicted in Figure 1, is presented to facilitate a detailed explanation.

In this figure, we explain the detailed architecture of the proposed MOGIC framework. The input query is "gummy candy", and the expected label is "Jelly Bean". The overall framework comprises of 4 different parts: Disciple, task specific loss functions, guidance from the Oracle to the Disciple, and the Oracle itself. The framework comprises of 2 stages: Oracle training and Disciple training. The Oracle training leverages both the query-side as well as the label-side ground truth metadata. As shown in the figure, the query-side ground truth metadata includes 2 memory items: "Gummi candies" and "Candy". The label side ground truth metadata includes just one memory item in this case: "gelatin desserts". This rich metadata is concatenated with the query and the label in the text form to train the Oracle. In stage 2, we train the Disciple based on guidance from the Oracle. This stage 2 is illustrated in detail in the Figure 4. The Disciple denoted by the green box, consists of an encoder, memory bank and a combiner $\mathcal{C}$. At train time, the Disciple first encodes both the query as well as the label using the same encoder. Since the Disciple has to be low latency, it must involve late fusion. For late fusion, we need embedding representations of both the query as well as the metadata items. To obtain embedding of the metadata items, the query is also sent to the memory bank. The combiner actually performs the late fusion using cross attention and outputs an enhanced query representation. Of course while training the Disciple, we need to ensure that the task specific loss is minimized. This loss tries to maximize the similarity between the embedding of the label and the enhanced query representation, and is illustrated in the task part of the Figure 4. The guidance part of the Figure 4 shows how the guidance is passed on by the Oracle to the Disciple using 4 different loss functions. These loss functions try to maximize the similarity between (a) Oracle's query representation and the Disciple's query representation (b) Oracle's label representation and the Disciple's label representation (c) Oracle's query representation and the Disciple's label representation, and (d) Oracle's label representation and the Disciple's query representation.

## N   ETHICAL CONSIDERATIONS

Our usage of data and terms of providing service to people around the world has been approved by our legal and ethical boards. In terms of social relevance, our research is helping millions of people find the goods and services that they are looking for online with increased efficiency and a significantly improved user experience. This facilitates purchase and delivery without any physical contact which is important given today's social constraints. Furthermore, our research is increasing the revenue of many small and medium businesses including mom and pop stores while also helping them grow their market and reduce the cost of reaching new customers.

