# OpenReview forum: "MOGIC: METADATA-INFUSED ORACLE GUIDANCE FOR IMPROVED EXTREME CLASSIFICATION"
_ICLR.cc/2025/Conference — Submitted to ICLR 2025_

### Official Review · Reviewer_6gPj · 2024-11-02

**Soundness:** 3
**Presentation:** 2
**Contribution:** 2
**Rating:** 5
**Confidence:** 4

**Summary:**

This paper introduces an oracle guidance framework for enhancing extreme classification tasks by integrating metadata through early-fusion techniques. By distilling from the oracle model, the disciple model can generate high-quality embeddings while maintaining low inference latency. Experimental results on benchmark datasets show that MOGIC consistently enhances performance on XC task, surpassing state-of-the-art methods.

**Strengths:**

1.The proposed method finds an efficient way to distill the oracle model, enhancing the disciple model’s ability to embed extreme classification data while maintaining low inference latency.
2.The authors conducted thorough experiments to validate the effectiveness of the proposed method.
3.Comprehensive implementation and experiment details underscore the soundness and practical viability of the proposed method.

**Weaknesses:**

1.In robustness analysis section, the author mentioned the MOGIC framework is more robust to Oracle models. Further analysis should be conducted on this phenomenon to verify that the robustness stems from the proposed MOGIC framework rather than from the OAK method itself.
2.The theoretical analysis on the oracle-guided losses is unaligned with the experiment implementation. Please see question 2 for details.
3.The presentation of the paper could be improved. There are many writing errors in the paper that hinder understanding, e.g. the unfinished caption of table 10.

**Questions:**

1.Why did you only present results on the LF-WikiSeeAlsoTitles-320K dataset when discussing MOGIC's generalizability across different disciple models? Does MOGIC still demonstrate generalizability on other datasets?
2.In the theoretical proofs in Appendix A, the loss function is assumed to be a decomposable binary loss rather than the non-decomposable triplet loss. Does the conclusion hold under triplet loss circumstances? Additionally, the alignment loss defined in line 797 has an asymmetric form. If that is not a typographical error, why is the binary loss of (xi*, zi, yi) calculated inner the binary loss of (xi, zi*, yi)?

---

> ### Author Response · Authors · 2024-11-23
>
> *Thank you for your valuable feedback! Please find a detailed response to your questions below.*
>
>  - **Robustness of MOGIC:** Thank you for your suggestion. We performed further experiments based on your suggestion.
>      - We have now included competing approach (OAK) results also in Table 9. Introducing noise in fused metadata at inference  can lead to up to 20% reduction in accuracy of the oracle, since early-fusion models rely on high-quality metadata at the input unlike  the late-fusion-based MOGIC and OAK models. Thus, in short, the surprising robustness in OAK is due to the late fusion architecture.
>     - We attribute the lack of robustness in the oracle model when faced with missing/noisy metadata to the early-fusion framework. The transformer model, in this case, is reliant on high-quality metadata being provided to the input of the transformer layers, which may not occur due to poor-quality or inaccurate prediction of metadata.
>     - Thus, you are right, the robustness stems from the late fusion approach and not specifically from MOGIC. Thus, both MOGIC and OAK are robust.
>
>  - **On the theoretical analysis:** Thank you for your insightful questions. We address them below:
>
>     1. In the theoretical proofs in Appendix A, the loss function is assumed to be a decomposable binary loss rather than a non-decomposable triplet loss. Does the conclusion hold under triplet loss circumstances?
>     - **Response**: Upon careful re-consideration, we find that the conclusions hold under a Lipschitz-continuous triplet loss setting as well. Accordingly, in the revised manuscript, we have modified the theoretical analysis to fit the triplet loss, and request the reviewer to take a look! We also thank the reviewer for this question which prodded us to improve our theory.
>
>     2. Additionally, the alignment loss defined in line 797 has an asymmetric form. If that is not a typographical error, why is the binary loss of (xi*, zi, yi) calculated inner the binary loss of (xi, zi*, yi)?
>     - **Response**: Yes, this was indeed a typo. We apologize for this inadvertent error, and have rectified it in the revised manuscript.
>
> - **Table 9 caption**: Thanks for pointing out. We have repaired it now. We have also handled other typos across the paper.
>
> - **Generalization to other datasets:** We have now included experiments on the Amazon datasets (Appendix K of the revised Manuscript). We also summarize the results in the table below. We observe that MOGIC achieves a P@1 of  50.05, which is a 1.68 gain over the 48.36 reported by the best baselines OAK on LF-Amazon-131K, and a P@1 of  47.01, which is a 0.6 gain over the 46.42 reported by the best baselines, OAK on LF-AmazonTitles-131K. This validates the strong generalizability of the MOGIC algorithm.
>
> - **Generalization to other disciples:** MOGIC can generalize across both disciples and datasets. To validate this, we now include results on training MOGIC with the DEXA and NGAME disciples on the LF-AmazonTitles-131K datasets in Appendix E. The table below summarizes these results, wherein we observe that MOGIC demonstrates performance gains across both disciples and datasets.
>
> *MOGIC with different Disciples on LF-AmazonTitles-131K*
>
> | Model | P@1 | P@5 | N@5 | PSP@1 | PSP@5 |
> |---|---|---|---|---|---|
> | MOGIC(OAK)  | 47.01  | 22.40  | 49.51  | 40.62 | 50.33 |
> | OAK | 46.42 | 21.88 | 49.06 | 39.76 | 49.78 |
> | MOGIC(DEXA) | 45.43 | 21.70 | 48.49 | 39.91 | 49.95 |
> | DEXA | 44.47 | 21.34 | 47.65 | 39.25 | 49.08 |
> | MOGIC(NGAME) | 44.27 | 21.26 | 47.48 | 39.48 | 49.18 |
> | NGAME | 43.44 | 21.16 | 47.10 | 39.00 | 49.00 |

---

> > ### Comment · Reviewer_6gPj · 2024-11-25
> >
> > Thank you for your rebuttal. However, I will keep my rating unchanged.

---

> > > ### Author Response · Authors · 2024-11-26
> > > **Request for Reconsideration**
> > >
> > > Dear Reviewer 6gPj,
> > >
> > > We sincerely appreciate your continued engagement with our work. We believe that the revisions we've made have significantly strengthened the paper.
> > >
> > > We are genuinely surprised that our efforts have not led to a change in your rating. We would be grateful if you could provide more specific feedback on the areas where you feel the paper still falls short.
> > >
> > > Your insights will be invaluable as we continue to refine our research.
> > >
> > > Thank you once again for your time and consideration.
> > >
> > > Regards,
> > >
> > > Authors

---

### Official Review · Reviewer_iMJc · 2024-11-04

**Soundness:** 2
**Presentation:** 1
**Contribution:** 2
**Rating:** 5
**Confidence:** 3

**Summary:**

The paper's focus is Extreme Classification (XC) tasks, in which the goal is to achieve high-accuracy within low-latency constraints. To address these challenges, the authors introduce MOGIC, which is a novel approach for metadata-infused oracle guidance for XC tasks. MOGIC is a 2-step process: first, it trains an early-fusion oracle classifier with access to both query- and label- side ground-truth textual metadata; then this oracle is used to guide the training of any existing memory-based XC student model via regularization.

**Strengths:**

The motivating problem (XC) has clear practical applicability, and the proposed approach (MOGIC) appears to be novel. Due to the paper's barely-fair organization and poor writing (including many non-grammatical and/or ambiguous statements that are difficult & time-consuming to parse), it is difficult to judge the paper's likely significance and impact.

**Weaknesses:**

The paper has two main weakness: (1) it lacks any latency numbers for a problem (XC) in which latency is critical, and (2) it is extremely difficult to read and comprehend due to its poor writing (e.g., long, ambiguous sentences, and dozens of missing articles) and a no-better-than-fair organization. All of this makes it extremely challenging to judge the paper's contributions and expected significance/impact.

1. ABSTRACT
- given the centrality of latency in XC, please add a sentence that summarizes MOGIC's latency and how it compares to existing approaches
- whenever you make claims on accuracy gains (eg, line 24), please quantify the claim as follows: "MOGIC improves P@1 by X%, from Y% to Z%". To be informative, the X% value must be judged in the context of Y% and Z%

2. Introduction
- please intuitively define/introduce, ideally with an illustrative example, all the key concepts of the paper: early/late-stage fusion, metadata-infused oracle, memory-based/free models, query-/label-side ground-truth, etc
- please restructure and re-write this section along the lines of the one in [Mohan et al, 2024]. That paper's introduction is easy to read and assimilate, and, as such, a great example to emulate. After I read Mohan's intro, it became much easier to understand yours.
- due to the odd formatting of the first column in Table 1, you should improve its readability by drawing the horizontal lines for each row. You should also (i) add the relevant query- and label- side meta-data [see lines 88-89], (ii) intuitively explain (for a general audience) the entire process, rather then relying  on the abstract row-names for rows 2-4, and (iii) explain the reason behind the mistakes of the the four approaches, with an emphasis on OAK, Oracle, and MOGIC (in particular, why the errors of MOGIC are disjoint from those of the Oracle?)
- ideally, Table 1 please should have an additional row for MOGIC( NGAME ), which is listed in the abstract as a major contribution.
- ideally, there should be an additional table (similar to Table 1) that intuitively explain the differences between the results with early- vs late- fusion
- lines 93-102 are just "tuning Table 1 into prose," without adding any insides or intuitions. As such they should be replaced insights/intuitions (or simply removed)
- lines 139-144: to increase the readability of this long, reference-rich sentence, you should replace the comma before "text-based" with a semi-column; then re-write the second part by bringing tabular data first, as it only was two references (vs 1+ lines of them)
- the caption of Table 2 is too long and too in-depth; move most of it the main narrative
- please replace the Low/High/VeryHigh labels in the last column by actual numbers; eg, O( ms ).
- Figure 1 is hard to read and interpret: (i) if the green box is "the architecture of the OAK disciple" then why does the horizontal-bracket labeled "Disciple" also covers the Encorder fed by the "Label: Jellybean" rectangle?, (ii)overall, you should spend a new paragraph that provides an intuitive, step-by-step explanation of what takes place in Fig 1.

3. Experimental Results
- in all tables, please use BOLD for the best result and UNDERLINED-ITALIC for the runner-up
- all tables/figures should include the results on all four datasets (not only a subset of them); at the very least, they should be added to the APPENDICES
- lines 338-342: provide references for both "the plethora of papers" and "the few of them that offer ground-truth data"
- Table 3: please provide the details on how EXACTLY you computed all values in the last four columns; for example, you have an "Avg Q/L" of 2.11, but 693K/312K = 2.22; similarly, in the first row "the nmb of memory items is smaller than the nmb of training queries", but it is larger in the second row; please explain why?
- for each of the four datasets, please select-and-justify a reasonably-ambitious target-latency (eg, answering N queries per second); then create a table (similar to Table 4) in which you compute the actual latencies for all the various approaches
- Table 4: please discuss the results where MOGIC is outperformed, especially when it loses to ANCE by 7.4% (50.99% vs 43.60); similarly, why does OAK outperform MOGIC on the same metric/dataset?
- Table 5: how comes that, by itself, DistilBERT heavily outperforms the other two oracles, but within MOGIC the differences are minimal?
- line 460: please quantify "more powerful and larger oracles;" the reader should have immediate access to this info in your paper
- the entire 4.2 section reads like "a bag of tables and results;" please re-organize it to emphasize the main results (eg, as 4..2.1, 4.2.2, etc)


- last but not least, in the robustness analysis
  (i) please show numbers for MOGIC's competing approaches, too
  (ii) please discuss in depth the sources/hypotheses for this robustness: is there any redundancy in the data sources? it is highly-counterintuitive that using only 40% of the data barely impacts MOGIC, while significantly impacting the Oracle (Table 9)
  (iii) same fore noise: how could be MOGIC barely impacted if 60% of the data is incorrect? also, when the impact on the oracle is greater than 50%, what factors contribute to "fading the impact of noise" on MOGIC by about an order of magnitude?

**Questions:**

- please section above

---

> ### Author Response · Authors · 2024-11-23
>
> *We thank the reviewer for their time and feedback. We are confident that your feedback has helped us improve the paper quality.*
>
> - **On writing clarity:** We would like to thank you for your detailed feedback in addressing the presentation clarity of the submission. We have incorporated the various changes suggested to improve the readability of the revised manuscript, including improving the clarity of the abstract, improving and restructuring Sec. 1, and providing more details on Figure 1. We hope that the revision addresses your concerns.
>
> - **MOGIC's latency in abstract:** We have now added a clarifying sentence in the abstract. More discussion about latency is below in this rebuttal.
>
> - **please intuitively define/introduce, ideally with an illustrative example, all the key concepts of the paper: early/late-stage fusion, metadata-infused oracle, memory-based/free models, query-/label-side ground-truth, etc:** We provide these definitions in Appendix L of the revised paper.
>
> - **restructure and re-write intro:** We have done multiple changes to the intro. We have also added additional Appendices (B, D, L and M) for further clarity.
>
> - **Table 1 formatting problems and detailed explanation:** Thank you for your suggestions. We have repaired the table formatting as suggested. We have also added a very detailed walk-through of an example from Table 1 in Appendix B. This appendix very clearly explains the query, label, metadata, and also predictions from various systems. We also detailed "what makes MOGIC(OAK) perform better than OAK?" in Appendix B.
>
> - **Table 1 please should have an additional row for MOGIC( NGAME):** Thanks for pointing this out. We have now added this row.
>
> - **there should be an additional table (like Table 1) that intuitively explain the differences between the results with early- vs late- fusion:** We have now included detailed explanation of early-fusion vs late-fusion in Appendix L. Also, please note that Table 1 already has early- and late-fusion examples. Oracle is early-fusion model. OAK and MOGIC(OAK) are late-fusion model examples.
>
> - **lines 93-102 (98-105 for revised paper) are just "tuning Table 1 into prose:**
>      - As suggested, we have now also added a very detailed walk-through of an example from Table 1 in Appendix B.
>      - OAK is trained with ground truth metadata and has no feedback available for it to know which metadata is useful. MOGIC(OAK) solves this issue, by regularizing the OAK using an Oracle model. Now difference in MOGIC(OAK) and Oracle's predictions can be attributed to the fact that, Oracle is more powerful and accurate model, therefore it is even able predict labels which were missing in ground truth (cf. Table 1). MOGIC(OAK), uses Oracle to regularize OAK making it more robust to noise as compared to OAK therefore making better prediction as compared to OAK but not able to outperform Oracle. This is also shows in Table 6 where Oracle is 8.16 points more accurate than MOGIC(OAK) which is 0.91 points more accurate that OAK at P@1 on LF-WikiSeeAlsoTitles-320K. i.e. Oracle > MOGIC(OAK) > OAK.
>
> - **lines 139-144 (146-149 of the revised paper):** We have reworded this sentence now.
>
> - **Low/High/VeryHigh labels in the last column of Table 2 (removed in revised paper):** Unfortunately, all of these methods have not been benchmarked for latency on a single dataset. Hence, we provided  Low/High/VeryHigh labels in Table 2 based on intuitive understanding of the mechanisms that these models use.
>      - We have classified "Retrieval-interleaved generation (RIG)" and "Reinforcement Learning for Feedback (RL4F)" as very high latency because RIG involves interleaving the process of generating responses with real-time data retrieval. This means the model continuously fetches and integrates external data while generating text, which can significantly increase the time required to produce a complete response. RL4F involves a multi-agent framework where a critique generator provides feedback to improve the outputs of a larger model like GPT-3. This iterative process of generating feedback and revising outputs can be time-consuming.
>      - We have classified "Retrieval-augmented generation", "Unified RAG (URAG)", "GRIT-LM", "MOGIC Oracle (Ours)" as high latency because these methods perform early fusion in the text space. They all add metadata in the text space thereby increasing the context length. Inference latency of Transformer models increases with increase in context length. Hence, these group of methods have higher latency compared to the methods which perform late fusion.
>      - We have classified OAK, DEXA, "MOGIC (OAK)" as low latency because these methods perform late fusion in the embedding space. This means that the context length is very short. Shorter context length helps avoid large latency.

---

> ### Author Response · Authors · 2024-11-23
>
> - **Figure 1 is hard to read and interpret: (i) if the green box is "the architecture of the OAK disciple" then why does the horizontal-bracket labeled "Disciple" also covers the Encorder fed by the "Label: Jellybean" rectangle?, (ii)overall, you should spend a new paragraph that provides an intuitive, step-by-step explanation of what takes place in Fig 1.**
>
>     - Thank for your attention to the detail about the green box in the figure. We have changed the green box to cover the entire disciple now.
>     - Here is a more detailed explanation for this figure: In this figure, we explain the detailed architecture of the proposed MOGIC framework. The input query is "gummy candy", and the expected label is "Jelly Bean". The overall framework comprises of 4 different parts: disciple, task specific loss functions, guidance from the Oracle to the disciple, and the Oracle itself. The framework comprises of 2 stages: Oracle training and disciple training. The Oracle training leverages both the query side as well as the label side ground truth metadata. As shown in the figure, the query side ground truth metadata includes 2 memory items: "gummi candies" and "Candy". The label side ground truth metadata includes just one memory item in this case: "gelatin desserts". This rich metadata is concatenated with the query in the text form to train the Oracle. In stage 2, we train the disciple based on guidance from the Oracle. This stage 2 is illustrated in detail in the figure. The disciple denoted by the green box, consists of an encoder, memory bank and a combiner C. At train time, the disciple first encodes both the query as well as the label using the same encoder. Since the disciple has to be low latency, it must involve late fusion. For late fusion, we need embedding representations of both the query as well as the metadata items. To obtain embedding of the metadata items, the query is also sent to the memory bank. The combiner actually performs the late fusion using cross attention and outputs an enhanced query representation. Of course while training the disciple, we need to ensure that the task specific loss is minimised. This loss tries to maximise the similarity between the embedding of the label and the enhanced query representation, and is illustrated in the task part of the figure. The guidance part of the figure shows how the guidance is passed on by the Oracle to the disciple using 4 different loss functions. These loss functions try to maximise the similarity between (a) Oracle’s query representation and the disciple’s query representation (b) Oracle’s label representation and the disciple’s label representation \(c) Oracle’s query representation and the disciple’s label representation, and (d) Oracle’s label representation and the disciple’s query representation.
>
> - **in all tables, please use BOLD for the best result and UNDERLINED-ITALIC for the runner-up:** As suggested, we use this formatting now for all the tables.
>
> - **all tables/figures should include the results on all four datasets:** Performing experiments on all datasets will take time. But we have already included a few of them in the revised draft. Ablation study in Table 5 now has results for LF-WikiTitles-500K dataset also besides the LF-WikiSeeAlsoTitles-320K dataset. We are currently running ablations on the full-text datasets LF-WikiSeeAlso-320K and LF-Wikipedia-500K which take 88 hours and 180 hours, respectively. Also, similar to the results for LF-WikiSeeAlsoTitles-320K dataset in Table 7, we have included results for LF-AmazonTitles-131K dataset in Appendix E. The training time for MOGIC(OAK) and Oracle on different datasets can be found in Appendix D.
>
> - **lines 338-342 (line 323 of the revised paper): provide references for both "the plethora of papers" and "the few of them that offer ground-truth data"**: Added these now.
>
> - **Table 2: Dataset statistics computation details**: (a) The Avg. queries/label is the average value of the number of positive labels associated with each query in the dataset. Similarly, to find the Avg. labels/query, given a label, we identify the number of queries for which that label is a positive, and compute the average of those numbers. Therefore, we have not computed *"Avg Q/L"* as 693K/312K*. Instead, we compute number of relevant labels for each query and then take the average over all the queries in the dataset. Thus, we have computed Micro-averages. We have now explained this clearly in the Table 2 caption. (b) As we already describe in Section 4.1, for WikiSeeAlso tasks, the Wikipedia categories that these articles are tagged with are used as metadata. Similarly, for the LF-WikiTitles-500K and LF-Wikipedia-500K tasks, the Wikipedia article titles connected to each original page via hyperlinks in the article are used as metadata. Therefore, there are fewer number of metadata points for the WikiSeeAlso tasks.

---

> ### Author Response · Authors · 2024-11-23
>
> - **for each of the four datasets, please select-and-justify a reasonably-ambitious target-latency (eg, answering N queries per second); then create a table (similar to Table 4) in which you compute the actual latencies for all the various approaches:**
>     - We consider the latency of a 6-layer (DistilBERT) dual encoder as our target latency since that is the current deployed system. Accordingly, we present target latency numbers below. We also present Prediction latency (in ms) of MOGIC(OAK) and DistilBERT Oracle.
>     - We provide inference latency of MOGIC(OAK) for different datasets on CPU, 8 threads, single query, 2 metadata vectors considering a PyTorch implementation. For full text datasets, since the query size is large by itself, the overall size of early concatenated text input for the Oracle is similar to the size of the input for the disciple. Hence, inference times are almost similar for disciple and Oracle for full-text datasets.
>     - As can be observed from the below tables, MOGIC(OAK) helps us achieve latency values close to our target while the Oracle is much slower on short text datasets.
>     - We now include an efficiency analysis for MOGIC, which we report in Appendix D of the revised submission.
>
>
> *Inference time (in ms) of DistilBERT, our target timings:*
>
> | Dataset/Models               | DistilBERT |
> |-|-|
> | LF-WikiSeeAlsoTitles-320K    | 8.62 |
> | LF-WikiTitles-500K           | 7.36 |
> | LF-WikiSeeAlso-320K          | 48.91 |
> | LF-Wikipedia-500K            | 46.31 |
>
> *Prediction latency (in ms) of MOGIC(OAK) and DistilBERT Oracle:*
>  Dataset/Models               | MOGIC(OAK) | DistilBERT Oracle
> -|-|-
>  LF-WikiSeeAlsoTitles-320K    |14.06|24.7
>  LF-WikiTitles-500K           |13.72|26.79
>  LF-WikiSeeAlso-320K          |52.85|49.56
>  LF-Wikipedia-500K            |50.27|48.88
>
> - **Table 4 (Table 3 of revised paper): please discuss the results where MOGIC is outperformed, especially when it loses to ANCE by 7.4% (50.99% vs 43.60); similarly, why does OAK outperform MOGIC on the same metric/dataset?**
>     - In extreme classification, methods that prioritize head labels (common labels) will have high precision but potentially lower propensity scores (ability to identify less frequent labels). Conversely, methods focusing on tail labels might exhibit high PSP but lower precision. Typically embedding based methods are accurate on tail labels and classifier-based methods are accurate on head labels. Effective extreme classification methods must strike a balance between these metrics, which is a core challenge in the field. The difference between method like ANCE and MOGIC(OAK) are the metadata free parameters. The observed drop in PSP scores with a rise in precision after applying classifiers in MOGIC(OAK) suggests that the model is focusing more on accurately classifying head labels while maintaining some capability for tail labels. This highlights the inherent tension in extreme classification and the importance of considering both precision and propensity scores for evaluation.
>     - MOGIC demonstrates marginal performance gains on long-text datasets such as LF-Wikipedia-500K. MOGIC demonstrates significantly higher gains on short-text datasets, where additional metadata-based context is more important. Although MOGIC outperforms baselines on both short-text (LF-WikiSeeAlsoTitles-320K and LF-WikiTitles-500K) and long-text (LF-WikiSeeAlso-320K and LF-Wikipedia-500K) datasets on most metrics, the relative gains on short-text applications are higher.
>
> - **Table 5 (Table 4 of revised paper): how comes that, by itself, DistilBERT heavily outperforms the other two oracles, but within MOGIC the differences are minimal?**
>     - **DistilBERT versus pre-trained oracles**: The LLaMA-2 and Phi-2 oracles are decoder-only generative models, where the model predicts next token given a context. These models are not trained to come up with the representation of the input text. As a heuristic we use the last token of the input prompt as the representation of the input and use that to train MOGIC and compute the statistics. On the other hand, the DistilBERT oracle is encoder representation model, pre-trained for sentence representation task and therefore outperforms the decoder-only generative oracles on the particular task. Nevertheless, we show that, via the MOGIC framework, the disciples are capable of leveraging the oracles’ signals to improve task performance.
>
> - **line 460(now 375): please quantify "more powerful and larger oracles;" the reader should have immediate access to this info in your paper**
>     - Thank you for your suggestion. We have now added parameter sizes for all 3 Oracles in Table 4.

---

> ### Author Response · Authors · 2024-11-23
>
> - **re-organize Section 4.2**
>     - We would have liked to create sub-sections. But due to lack of space, we have organized Section 4.2 using paragraphs titled with crisp bold headings. Further, we have now aligned the table caption also with the same headings so that it is easier for the reader to link the table with the right para (subsection). Hope that helps understand the motivation for each experiments clearly.
>
> - **in the robustness analysis (i) please show numbers for MOGIC's competing approaches, too (ii) please discuss in depth the sources/hypotheses for this robustness...**
>     - We have now included competing approach (OAK) results also in Table 9. Introducing noise in fused metadata at inference  can lead to up to 20% reduction in accuracy of the oracle, since early-fusion models rely on high-quality metadata at the input unlike  the late-fusion-based MOGIC model and OAK models. Thus, in short, the surprising robustness in OAK is due to the late fusion architecture.
>     - We attribute the lack of robustness in the oracle model when faced with missing/noisy metadata to the early-fusion framework. The transformer model, in this case, is reliant on high-quality metadata being provided to the input of the transformer layers, which may not occur due to poor-quality or inaccurate prediction of metadata.
>
>
> - **Label-side metadata for the examples:** We have already provided query side metadata in the row *"Ground-truth Query Metadata"*
>
> |Query | Grass court | Tangbe |
> |---|---|---|
> |Ground truth labels | Clay court, Carpet court, Hardcourt | Mustang District, Kali Gandaki Gorge, Kali Gandaki River, Upper Mustang, Gandaki River|
> | Ground truth Label Metadata | **Clay court:** [Tennis terminology, Sports rules and regulations, Clay, Tennis court surfaces], **Carpet court:** [ Carpet court tennis tournaments, Tennis court surfaces], **Hardcourt:** [Tennis terminology, Tennis court surfaces] | **Mustang District:** [Mustang District, Districts of Nepal, Gandaki Pradesh, Populated places in Mustang District], **Kali Gandaki Gorge:** [Water gaps, Canyons and gorges of Nepal], **Kali Gandaki River:** [], **Upper Mustang:** [History of Nepal, Former monarchies of Asia, Unification of Nepal, Nepali-speaking countries and territories, Tibetan Buddhist places], **Gandaki River:** [Tributaries of the Ganges, International rivers of Asia, Rivers of Nepal, Rivers of Bihar, Rivers of Patna, Ancient Indian rivers]|

---

> > ### Comment · Reviewer_iMJc · 2024-11-25
> > **Thank you for the comprehensive rebuttal**
> >
> > Thank you for the comprehensive rebuttal: based on all four reviews & your respective rebuttals, I will maintain my current Rating

---

> > > ### Author Response · Authors · 2024-11-26
> > > **Request for Reconsideration**
> > >
> > > Dear Reviewer iMJc,
> > >
> > > We have thoroughly addressed all 22 points, including additional experiments to validate our claims, resulting in significant improvements to the paper’s clarity and rigor.
> > >
> > > We kindly request that you reconsider your rating in light of these revisions and are happy to provide further clarification if needed.
> > >
> > > Thank you for your time and expertise.
> > >
> > > Regards,
> > >
> > > Authors

---

### Official Review · Reviewer_U9Gh · 2024-11-04

**Soundness:** 3
**Presentation:** 2
**Contribution:** 2
**Rating:** 5
**Confidence:** 2

**Summary:**

This work proposes MOGIC, a method for achieving high accuracy, low latency extreme classification (XC). In MOGIC, the authors first train an expensive early-fusion oracle classifier that can access metadata as text. Subsequently, this oracle is used to regularize the training of existing XC methods like OAK. This consistently improves quality by a couple of percentage points when applied over a variety of XC methods, including OAK, DEXA, and NGAME, on a few XC datasets.

**Strengths:**

1. The authors conduct evaluation on several benchmarks and by applying MOGIC over three different XC baselines, and show consistent gains across the board. This suggests the method has some fundamental additive capacity to add to the field of XC.

2. The proposed method balances quality and cost, while boosting quality over baselines. This type of improvement is encouraging to see.

**Weaknesses:**

1. The paper is not particularly easy to follow. In particular, while I appreciate the potential generality of the proposed framework, the current presentation comes at the cost of concreteness. For one, the paper needs an end-to-end example of how different components interact with a given query at training and inference time, e.g. how OAK works and how OAK via MOGIC works. In general, the descriptions of the task and the oracle are a lot more complicated than I think they need to be.

2. Building on #1, the discussion of the oracle, saying things like "Oracle leads to very high accuracy on the downstream XC task but is computationally expensive to deploy. It entails too high inference times for any real world application, due to the large context length" is quite perplexing. An "oracle" suggests to me access to privileged information, generally not available at test time; if so, the computational cost is the least of anyone's concerns for deploying the oracle. (I imagine I simply do not fully understand this section!) When the authors discuss presenting the "Labels" to the oracle, I'm left unsure if they mean concatenating all 312,000 labels (is this why the context is so long?) or the ground truth labels (why is that long, in that case?). Overall, the discussion of the oracle and the overall pipeline is fairly opaque.

2. Given the increased training-time cost, and the complexity of the method (at least as currently presented notationally, see weakness #1), gains of 1-2 percentage points with a fairly standard intuition (i.e., distilling an expensive oracle) may not be the most rewarding tradeoff, in a way that weakness the core contribution of this work.

**Questions:**

1. The authors write "XC tasks involve sparse query representation, and are short-text in nature". Is this key to the contribution here? It just seems like an inaccurate/overly general statement, e.g. see BioDEX as a fairly standard XC task where the queries are anything but short.

2. The paper says "discipline" 3 times. Is this a typo? Is it supposed to be "disciple"?

---

> ### Author Response · Authors · 2024-11-23
>
> We sincerely thank you for your valuable feedback! Please find a detailed response to your questions below.
>
>  - **Improving paper writing**: Thank you for the suggestions. We have incorporated the flow of an end-to-end example in Appendix B to improve readability of the paper and we hope that you find the presentation of the revised manuscript to be clearer. We have also corrected the typos.
>
> - **End-to-end example:** We now also walk through an example for MOGIC in Appendix B. Consider the query “Grass Court” (subsequently referred to as Q) with a ground-truth lable “Clay Court” (L) The follwing three steps are followed by the query:
>     1. **Query Processing Block:** involves the following steps
>         - *Metadata Retrieval:* Q is sent to the memory bank to retrieve relevant metadata (and its vector representations) such as “Courts by type” or “Landforms”, and “Grasslands”.
>         - *Query Encoding:* Q is passed through the encoder to obtain its vector representation.
>         - *Query Enrichment:* The query representation is fused with the metadata representation using a cross-attention layer to create an enriched query representation.
>     2. **Label Processing block:** similar to the above query processing
>         - *Label Encoding:* L is passed through the encoder to obtain its vector representation. This encoder is shared between query and the label.
>         - *Label Enrichment:* This representation is further enriched by combining it with a separate free parameter.
>     3. **Oracle representation block:** Q is concatenated with its associated metadata "Tennis terminology", "Sports rules and regulations", "Tennis court surfaces" to form an enriched query "Clay Court, Tennis terminology, Sports rules and regulations, Tennis court surfaces". This enriched query is passed through the oracle encoder to obtain its vector representation. Similarly "Clay court" is concatenated with "Clay", "Tennis court surfaces" and "Clay tournaments" and passed to the shared oracle encoder to obtain its vector representation.
>
>     The above blocks are then used for training and inference,
>     1. **Inference:** During inference for the query "Grass Court" we compute its query representation using the *Query processing block* and then calculate its similarity with all label representations computed from the *Label processing block* in the dataset, including "Alabama", "Clay Court", "Carpet Court", "Hardcourt" and "Henry Moore", to determine their relevance to the query using cosine similarity distance.
>     2. **Training:** Unlike inference, training uses all the blocks and involves the following steps:
>         - Vector Representations: We compute query and label representations for both Q and L using the *Query processing block* and the *Label processing block*.
>         - Triplet Loss: We then apply triplet loss to the query and label representations, as is common in retrieval methods.
>         - Oracle Regularization: To further regularize the Disciple model, we introduce additional triplet loss terms (a) between the query "Grass Court" Oracle representation and the "Clay Court" label representation. (b) between the query "Clay Court" Oracle representation and the "Grass Court" label representation. (c\) mean squared error (MSE) loss to minimize the distance between the "Grass Court" oracle representation and its query representation and the "Clay Court" oracle representation and its label representation.
>
> - **Clarification on the Statement “Oracle leads to very high accuracy …, due to the large context length”:** We agree with the Reviewer that the oracle has access to privileged information (*i.e.,* ground-truth textual metadata), which is not available at test time to the disciple, which is a contributing factor to the high accuracy of the Oracle. However, beyond this, as we note in Sec. 1, the computational cost is also crucial in real-world applications, wherein text-based early-fusion models incur higher inference costs due to relatively larger context lengths, unlike embedding-based late-fusion models. We have included this discussion in the revised manuscript (Line 254) to improve the clarity of the aforementioned sentence. And now include an efficiency analysis for MOGIC and Oracle, which we report in Appendix D of the revised submission.

---

> ### Author Response · Authors · 2024-11-23
>
> - **Oracle Sequence length in MOGIC:**
>      - Please note that metadata is different from labels. "on sponsored search ads task that involves query-to-ad-keyword prediction, the query-side metadata is obtained by mining the organic search webpage titles clicked in response to the query on the search engine, while on the Wikipedia categories prediction task, other Wikipedia article titles connected to the original page via hyperlinks could serve as the metadata. " as mentioned in lines 46-50 of our submission.
>      - None of the 312,000 labels are concatenated to the input. We concatenate the text of the query and the metadata (which is different from the labels) associated with the query. The number of metadata concatenated is dependent on the datasets. In LF-WikiSeeAlsoTitles-320K and LF-WikiSeeAlso-320K there are 4.89 metadata (category) per query on average (cf. Table 2) and in LF-WikiTitles-500K and LF-Wikipedia-500K there are 15.95 metadata (hyperlinks) per query on average. For example, on the LF-WikiSeeAlsoTitles-320K dataset, given a query "Grass court" and its metadata (in this case exactly three) "Courts by type", "Landforms" and "Grassland", we form a super query by concatenating all the text "Grass court Courts by type Landforms Grassland".
>
> - **Method complexity versus performance tradeoff**:
>      - As we already note (cf. L363-365 of the revision), MOGIC is merely a regularization framework, and we observe that inference times for a given baseline disciple model and its MOGIC variant are identical. There is slight increase in the training time due to computation of the additional regularization terms in the loss function.
>      - As seen from the examples in Table 1 of the submission, the 2% performance gains are coupled with improved and diverse predictions for the query, and the ability to retain the original intent of the query, by overcoming the noise in the metadata. As we note in Lines 98-105, this potential is unlocked by the MOGIC framework. Furthermore, as we show in Tables 8 and 9, the performance gains are dependent on metadata quality and quantity. Table 8 also shows performance variations w.r.t amount of metadata and Table 9 shows the effects of noise in meta-data. If precision and diversity of the metadata can be improved in the future, the performance gains achieved by MOGIC can be increased further.
>      - Also note that 1-2% gains are very significant in the computational ads business. They can lead to millions of dollars of revenue impact every day.
>
>
> - **Short-text nature of XC**: We agree that short-text XC tasks are not the only kind, but they are frequently encountered and important [1,2]. *That said, short text is not key to the success of MOGIC*. However, we also note that, on short-text datasets wherein additional metadata-based context is more important, MOGIC demonstrates significantly higher gains. The results present in Table 3 validate this, wherein MOGIC outperforms baselines on both short-text (LF-WikiSeeAlsoTitles-320K and LF-WikiTitles-500K) and long-text (LF-WikiSeeAlso-320K and LF-Wikipedia-500K) datasets, but the relative gains on short-text applications are higher **(1.74% improvements of P@1 on short text, compared to 0.58% improvements of P@1 on long text)**.
>
> [1] K. Dahiya et al., DeepXML, WSDM 2021
>
> [2] A. Mittal et al., DECAF, WSDM 2021

---

> > ### Comment · Reviewer_U9Gh · 2024-11-25
> >
> > Thanks for the response. I will keep my scores.

---

> > > ### Author Response · Authors · 2024-11-25
> > >
> > > Thanks for acknowledging our response.
> > > We would like to understand if the reviewer has further concerns. Else, we request the reviewer to kindly reconsider the scores if their concerns have been addressed.

---

### Official Review · Reviewer_4a5S · 2024-11-05

**Soundness:** 3
**Presentation:** 2
**Contribution:** 3
**Rating:** 6
**Confidence:** 5

**Summary:**

In this paper, the authors propose MOGIC to leverage an additional oracle model and metadata for extreme classification. In the first phase, an oracle model for early fusion is trained with metadata. Then a smaller encoder with the same architecture as OAK is trained by distilling the oracle model with auxiliary losses in the second stage. The experiments are conducted on some benchmark datasets based on Wiki. The experimental results show that the MOGIC framework can improve OAK across all datasets. The authors also conducted several studies to demonstrate the effectiveness of each component. Besides, there is also a theoretical analysis of the loss used in the distillation.

**Strengths:**

* Great improvements over different base models.
* Both alignment and matching to approach the Oracle model are helpful.
* Good theoretical analysis for the framework and optimization losses.

**Weaknesses:**

* The experiments only use the Wiki datasets, so the framework is unproven for other domains, such as e-commerce like Amazon datasets.
* Performance gaps between baselines and MOGIC are unclear about whether they are from the framework, pre-trained oracle model, or ground-truth metadata in training.
* Lack of reports and analysis on training and inference time when the authors emphasize the efficiency.
* Writing and organization can be improved.

**Questions:**

* I wonder whether the two-stage framework is needed when it could be feasible to train models directly with "ground-truth" metadata (and, of course, use the predicted metadata during inference). It would be great to patch an experiment to demonstrate the benefit of this two-stage design.
* I guess PCA is one of the main reasons why Phi-2 and LLaMA-2-7b underperform DistiBERT as oracle models, especially the authors do not provide more details. I would suggest also reporting the performance without PCA, even if it would take a longer time.
* It would be great to patch an efficiency analysis for both training and inference.
* I would also suggest polishing the writing and organization to ease the reading. For instance, the concept of predicted and ground-truth metadata are not explained until the experiment section, but the term ground-truth metadata is used throughout the whole paper.

**Details Of Ethics Concerns:**

I did not see any flag of ethics concerns.

---

> ### Author Response · Authors · 2024-11-23
>
> *We thank the reviewer for their strong positive, insightful and valuable comments and suggestions which are crucial for further strengthening our manuscript.*
>
>  - **Experiments on Additional datasets:** We have now included experiments on the Amazon datasets (Appendix K of the revised Manuscript). We also summarize the results in the table below. We observe that MOGIC achieves a P@1 of  50.05, which is a 1.68 gain over the 48.36 reported by the best baseline OAK on LF-Amazon-131K. MOGIC also achieves a P@1 of  47.01, which is a 0.6 gain over the 46.42 reported by the best baseline, OAK on LF-AmazonTitles-131K. This validates the strong generalizability of the MOGIC algorithm.
>
> *Performance on LF-Amazon-131K:*
> | Models | P@1  |P@5   | N@5   | PSP@1   | PSP@ 5 |
> |---|---|---|---|---|---|
> | MOGIC(OAK)  | 50.05  | 23.72  | 52.87  | 41.90 | 53.80 |
> | OAK  | 48.36  | 22.20 | 51.27  | 40.26   | 52.21|
>
>
> *Performance on LF-AmazonTitles-131K:*
> | Models | P@1  |P@5   | N@5   | PSP@1   | PSP@ 5 |
> |---|---|---|---|---|---|
> | MOGIC(OAK)  | 47.01  | 22.40  | 49.51  | 40.62 | 50.33 |
> | OAK  | 46.42  | 21.88  | 49.06  |39.76   | 49.78 |
>
> - **Performance gains of MOGIC:** The performance gains for MOGIC can be attributed to the novel algorithmic framework we propose. To validate this, we performed two extra ablations as described below (also included in Appendix C of the revised paper).
>     1. Training MOGIC(OAK) with a distilBERT Oracle trained without the metadata. Here, we observe that the choice of Oracle having access to metadata or auxiliary information significantly impacts the overall model performance.
>     2. Training the disciple OAK in MOGIC(OAK) with ground truth metadata. We observe that training the MOGIC disciple with predicted metadata improves performance. This is due to the potential mismatch between training and inference distributions when the disciple is trained solely with ground truth metadata. While the combiner can filter out unnecessary metadata, maintaining a closer training-inference distribution proves helpful. The table below summarizes the results:
>
> *Results on LF-WikiSeeAlsoTitles-320K:*
>
> | Models | P@1  |P@5   | N@5   | PSP@1   | PSP@ 5 |
> |---|---|---|---|---|---|
> | MOGIC (OAK)  | 34.62  | 17.93 | 27.44  | 35.70   | 33.18 |
> | MOGIC(OAK) + Oracle w/o Metadata  | 34.09  | 17.43  | 26.95  | 34.9 | 32.13 |
> | MOGIC(OAK) on 'OAK + ground truth Metadata'   |34.25  | 17.75  | 26.81  | 35.45 | 32.64 |
> | OAK | 33.71  | 17.12  | 24.53  | 33.83 | 30.83 |

---

> ### Author Response · Authors · 2024-11-23
>
> - **Training and inference time:** We now also report the training and inference time and comparisons between DistilBERT Oracle and MOGIC(OAK). The table below summarizes these numbers. As we already note in Section 4.2 of the submission (Line 363-365 of the revision), MOGIC is merely a regularization framework, and we observe that inference times for a given baseline Disciple model and its MOGIC variant are identical. There is marginal increase in the training time due to computation of the additional regularization terms in the loss function. We summarize the times below.
>     1. **Training**: All models were trained on 4 AMD MI200 GPUs. MOGIC(OAK) uses a context length of 32 for short-text datasets and 256 for full-text datasets. The Oracle is trained with context length of 128 and 256 for short and full-text datasets, respectively. The training time for distilBERT Oracle is roughly 2x that of the MOGIC(OAK) due to the increased context length for short-text datasets. We attribute the higher training time for the Oracle (for short text datasets) to its longer context length leading to larger attention matrices used to calculate backward-pass gradients.
>     2. **Inference**: We provide inference latency of MOGIC(OAK) for different datasets on CPU, 8 threads, single query, 2 metadata vectors considering a PyTorch implementation. For short text datasets, MOGIC(OAK) is much faster. For full text datasets, since the query size is large by itself, the overall size of early concatenated text input for the Oracle is similar to the size of the input for the Disciple. Hence, inference times are almost similar for Disciple and Oracle for full-text datasets.
>     3. We also provide end-to-end production latency to validate real-world feasibility, with inference on CPU, with single query, and two metadata vectors. Model optimizations allow for much smaller latencies, reducing the gap between OAK encoder and MOGIC(OAK) inference latency - making the latter feasible. p95 is 95th percentile latency, p99 denotes the 99th percentile latency, p999 denotes the 99.9th percentile latency in the table below. (cf. Appendix D of the revised paper)
>
> *Training and prediction time of MOGIC(OAK) and DistilBERT Oracle:*
> | Dataset/Models               | MOGIC(OAK) | MOGIC(OAK) | DistilBERT Oracle   | DistilBERT Oracle |
> |-|-|-|-|-|
> |             |Train time (hrs) |Pred. time (ms)| Train time (hrs) | Pred. time (hrs) |
> | LF-WikiSeeAlsoTitles-320K    | 25 | 14.06   | 45 | 24.70 |
> | LF-WikiTitles-500K           | 41 | 13.72   | 69 | 26.79 |
> | LF-WikiSeeAlso-320K          | 88 | 52.85   | 90 |  49.56|
> | LF-Wikipedia-500K            | 180 | 50.27   | 173 | 48.88 |
> | LF-AmazonTitles-131K         | 8  |  13.66  | 13 | 25.21 |
> | LF-Amazon-131K               | 22 |  51.91  | 23 | 49.99 |
>
> *Real world deployment latency*
> | Algorithm | Mean | p95 | p99 | p999 |
> |---|---|---|---|---|
> | OAK encoder| 10.30 | 20.05 | 22.59 | 23.49 |
> | MOGIC(OAK)| 14.95 | 22.34 | 25.07 | 27.08 |
>
> - **Need for two-stage framework:**
>      - Note that the baseline disciple models were trained directly with "ground-truth" metadata. Thus, if we were to remove the Oracle, this would be identical to the baseline disciple model, which we already show, is outperformed by the corresponding MOGIC variant (cf. Tables 3, 7 and 13).
>      - We also performed more ablations for reducing the two-stage framework to a single stage, to further clarify the benefits of our approach, with results summarized in the table below (and also Appendix C of the revised paper).
>     1. **Perform early fusion for disciple with ground truth metadata**: As shown by Mohan et al., 2024 [1], early fusion can negatively impact performance due to noisy metadata confusing the model. (Table 8 in [1]). This model concatenates query and ground truth metadata text in the input during training just like our Oracle and uses predicted metadata text during inference.
>     2. **Perform both early and late fusion for disciple with ground truth metadata**: The model combines the early fusion of Oracle and late fusion of OAK into a single model. This also performs worse as the noisy metadata text in the input confuses the model.
>
> *Ablations for combining the two-stage framework*
> |Model Variant |P@1 | P@5 | N@5 | PSP@1 | PSP@5|
> |-|-|-|-|-|-|
> | Early Fusion (similar to REALM)| 28.49 | 14.52 | 29.56 | 22.26 | 26.52 |
> | MOGIC(OAK) + Early Fusion| 29.30 | 14.88 | 29.92 | 22.21 | 26.87 |

---

> ### Author Response · Authors · 2024-11-23
>
> - **Performance of larger oracles without PCA:** Using LLaMA/Phi as an oracle mandates the use of a down-projection layer, as the models operate with a 4096-dimensional space, while disciple models such as OAK operate in a 768-dimensional space. As an alternative to the PCA model, we now also consider a learnable projection transform atop the oracle. An updated analysis of oracles is presented in Appendix F of the revised manuscript. We observe that the 7B LoRA-fine-tuned LLaMA-2 oracle with a projection layer performs on par with the 66M parameter standard-fine-tuned DistilBERT oracle. This presents a performance *versus* training overhead trade-off between using small, but task-specific, oracles and large pre-trained, but fine-tuned, general purpose oracles, which is a promising direction for future research.
>
> *MOGIC on the LF-WikiSeeAlsoTitles-320K dataset with different oracle models and an addition projection layer*
> |Oracle Models| P@1 | P@5 | N@5 | PSP@1 | PSP@5|
> |-|-|-|-|-|-|
> |LlaMA-2 |34.64|17.93|35.71|27.28|33.02|
> |Phi-2|34.34|17.73|35.44|27.09|32.71|
>
> - **Efficiency analysis:** We now include an efficiency analysis for MOGIC, which we report in Appendix D of the revised submission. We also already answered this above.
>
> - **Polishing paper writing**: Thank you for the suggestions. We have improved the readability of the paper and we hope that you find the presentation of the revised manuscript clearer to read. Specifically Lines 74-75 now describe the concept of predicted and ground-truth metadata.

---

> > ### Author Response · Authors · 2024-11-26
> > **Eagerly waiting for feedback on revisions made**
> >
> > Dear Reviewer 4a5S,
> >
> > We have provided detailed explanations and additional experiments to address your concerns. We have also uploaded a revised manuscript with all new additions clearly highlighted in blue font. We are eager to receive your feedback on the implemented changes.
> >
> > Regards,
> >
> > Authors

---

### Author Response · Authors · 2024-11-23
**Summary of our responses and revision**

We would like to thank the various reviewers for taking effort to review our paper! Please find a detailed response to your questions below.

**We would like to highlight the work's strengths identified by the reviewers:**

1. Reviewer 4a5S notes the improvements over the various base models, and the value of including an alignment term to approach an Oracle model.
2. Reviewer 4a5S remarked on the good theoretical analysis presented for the oracle guidance framework.
3. Reviewer U9GH finds our method to show *consistent gains across the board*, having a *fundamental additive capacity to add to the field of XC*
4. Reviewer U9GH also notes that our method balances quality and cost, while boosting quality over baselines, which is improvement that is encouraging to see.
5. Reviewer iMJc remarks positively about the novelty of our work
6. Reviewer 6gPj appreciates our thorough experimentation, and the effectiveness of our proposed approach, and find *practical viability* in the MOGIC method.

We have updated the main manuscript and the appendix to address these following comments. **The changes made in the manuscript are highlighted in blue color.**

**To summarise the changes we have incorporated as a result of the feedback received from the reviewers, we have**
1. Included additional results on the Amazon dataset in Appendix K. We show that MOGIC achieves a P@1 of  50.05 on LF-Amazon-131K, and a P@1 of 47.01 on LF-AmazonTitles-131K, outperforming the best-case baselines in both cases.
2. Included an extensive efficiency analysis, which is available in Appendix D of the revision.
3. Performed multiple ablations on (a) The source of performance gain for MOGIC (Appendix C), (b) Generalization of MOGIC across disciples (Appendic E); \(c) Oracle models with linear projection layers (Appendix F), and (d) Robustness of Noise in MOGIC (Table 10).
5. Improved the writing of the Introduction and included an end-to-end walkthrough of the MOGIC pipeline in Appendix B. We have also included a detailed discussion of Preliminaries in Appendix L.
6. Extended to theoretical analysis to also account for triplet-based losses, as considered in the experiment section of MOGIC in Appendix A.

---

### Author Response · Authors · 2024-11-25
**All concerns addressed**

Dear reviewers, we have addressed all the limitations/ concerns that we were asked (summarized below). The revised scores do not take this into account. Are there any more experiments or clarifications that are still not resolved -- we are happy to address them too.


|                            | **Main limitations / requests**                                                   | **Status**                                                                                                                                                                                                                                       |
|--------------------------------|-----------------------------------------------------------------------------------|---------------------------------------------------------------------------------------------------------------------------------------------------------------------------------------------------------------------------------------------------|
| **Generalization**             | Generalization to other domains: add an e-commerce dataset (reviewers 4a5S, 6gPj) | Added results on two Amazon based datasets - and MOGIC shows similar gains on this e-commerice domain                                                                                                                                             |
|                            | Generalization to other disciples and oracle variants (iMJc, 6gPj)                | Added results on various oracle variants that were requested (LLaMA, Phi with linear layer instead of PCA) and three disciples (showing gains across disciples across datasets).                                                                  |
| **Analysis**                   | Efficiency analysis (4a5S, iMJc)                                                  | Added inference latency (showing that our approach is faster than the Oracle model) and added training time information (quantifying the overhead for training task specific oracles).                                                       |
|                            | Robustness analysis (6gPj)                                                        | Added robustness study for OAK, showing that the robustness comes from the late fusion approach common to both MOGIC and OAK.                                                                                                                     |
|                            | Ablation analysis (4a5S, 6gPj)                                                    | Added all suggested ablations (regularization loss, training disciples with predicted metadata instead of ground truth, late infusion, and oracle's access to ground truth metadata), showing that all these components contribute to the gains.  |
| **Presentation issues/ proof** | Assumptions in Theory (6gPj)                                                      | Added updated theory, showing that the conclusions even hold under a Lipschitz-continuous triplet loss setting.                                                                                                                                   |
|                            | Writing and presentation (U9Gh, iMJc)                                             | Added updated paper, with preliminaries, end-to-end walkthrough examples, and updated sections as requested.                                                                                                                                      |

---

### Meta-Review · Area_Chair_8R5o · 2024-12-16

**Metareview:**

The paper presents an approach to incorporate additional metadata to enhance the performance of extreme classification. Rather than separately retrieving relevant metadata and then fusing with the standard model predictions, the proposal is to perform a form of distillation: one trains an oracle model based on the metadata, and then uses this model's embeddings to regularize those of the student ("disciple") model. The proposal is shown to yield consistent gains on extreme classification benchmarks.

The initial set of reviews offered a large number of concerns, including the lack of latency analysis, results being presented on only Wiki benchmarks, and a significant lack of clarity in several sections. The authors had a detailed response including more results and discussion. These involve significant changes to the original paper.

Following further discussion, reviewers were not convinced on the paper's technical or conceptual contribution being significant enough to warrant the complexity of the approach. From the AC's reading, one may indeed regard the proposed method as a particular instantiation of knowledge distillation, which the paper demonstrates can be useful for extreme classification. This is commendable, but the gains are not exceptional, the method is somewhat complex, and conceptually the finding is in keeping with the known efficacy of knowledge distillation. Given these, we believe it is challenging to accept the paper at present.

**Additional Comments On Reviewer Discussion:**

Reviewers raised a large number of concerns on the initial submission, starting from the lack of clarity in the presentation of key ideas, to the absence of a latency comparison to baselines. Following the response and revision, the authors provided a large number of changes to both the writing and experimental results. These are undoubtedly to the benefit of the paper, and appreciated for their thoroughness.

Reviewers however generally remained not in favor of acceptance, with two primary concerns remaining:

(1) the proposal not offering much new conceptual insight, being a particular instance of distillation albeit expressed in a challenging-to-follow manner, and

(2) the latency analysis being somewhat mixed, with the proposed method exceeding the oracle timings on long-text settings.

From the AC's reading, point (1) may be particularly relevant: while the paper is significantly improved in writing from the initial version, the body remains somewhat challenging to follow, and the payoff is a particular type of distillation method with reasonable but not exceptional gains.

---

### Decision · Program_Chairs · 2025-01-22

Reject